# GUIDING SKILL DISCOVERY WITH FOUNDATION MODELS

## ABSTRACT

Learning diverse skills without hand-crafted reward functions could potentially accelerate reinforcement learning in downstream tasks. However, existing skill discovery methods focus solely on maximizing the diversity of skills without considering human preferences, which leads to undesirable behaviors and possibly dangerous skills. For instance, a cheetah robot trained using previous methods learns to roll in all directions to maximize skill diversity, whereas we would prefer it to run without flipping or entering hazardous areas. In this work, we propose a **Fo**undation model **G**uided (FoG) skill discovery method, which incorporates human intentions into skill discovery through foundation models. Specifically, FoG extracts a score function from foundation models to evaluate states based on human intentions, assigning higher values to desirable states and lower to undesirable ones. These scores are then used to re-weight the rewards of skill discovery algorithms. By optimizing the re-weighted skill discovery rewards, FoG successfully learns to eliminate undesirable behaviors, such as flipping or rolling, and to avoid hazardous areas in both state-based and pixel-based tasks. Interestingly, we show that FoG can discover skills involving behaviors that are difficult to define. Interactive visualisations are available from
https://sites.google.com/view/iclr-fog.

## 1 INTRODUCTION

Reinforcement learning (RL) has shown promising results in robotics (Tang et al., 2024; Wu et al., 2023) and games (Vasco et al., 2024; Zhang et al., 2024). Typically, RL requires carefully designed reward functions, which demand significant expert effort (Schenck & Fox, 2018; Sowerby et al., 2022). In contrast, Unsupervised RL (URL) (Laskin et al., 2021; Rajeswar et al., 2023) aims to eliminate task-specific reward functions and train agents in a self-supervised manner. One key direction in URL is pre-training agents to acquire diverse skills that can potentially be useful in downstream tasks (Eysenbach et al., 2018; Park et al., 2023b), termed unsupervised skill discovery. Most prior methods in unsupervised skill discovery focus on maximizing skill diversity, encouraging agents to achieve diversity in both low-level behaviors and high-level policies. For instance, a cheetah robot trained using previous methods (Park et al., 2022; 2023b) learns to flip or roll (low-level behavior) in all directions (high-level policy). However, wide motions like flipping or rolling could damage the robot, and entering restricted areas might pose safety risks. Ideally, we want agents to learn skills that are not only diverse, but also align with specific intentions, such as eliminating undesirable behaviors or avoiding certain areas.

To integrate human intentions into skill discovery, Kim et al. (2024b) trains agents to align with behaviors presented in pre-collected demonstrations, enabling the discovery of diverse and desirable skills. However, collecting such demonstrations requires expert-level operations (Fu et al., 2024; Pertsch et al., 2021), which may not be feasible in tasks where human performance is limited. For instance, in high-dimensional humanoid robotic control tasks, we may want a humanoid robot to adopt stretched postures rather than twisted ones. Yet, defining criteria such as "stretched" or "twisted", which is necessary for designing a reward function to collect such demonstrations, is extremely challenging. Concurrent work by Rho et al. (2024) utilizes large language models (Achiam et al., 2023) to generate textual descriptions of every state based on state-specific queries. These descriptions are then embedded (Reimers, 2019) to form a reward function to guide agents to discover semantically diverse skills. However, this method only works in state-based tasks (language mod-

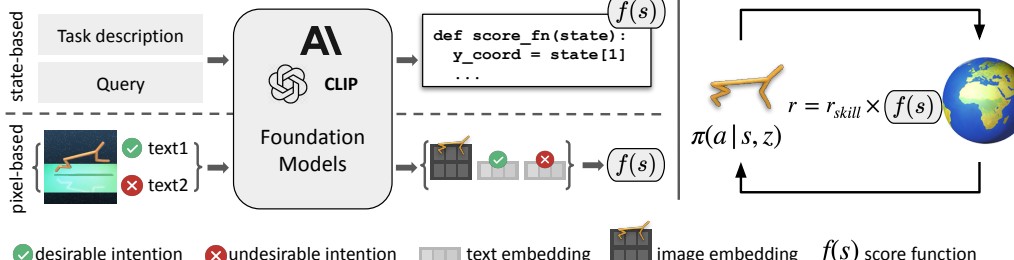

Figure 1: FoG leverages foundation models (such as ChatGPT, Claude and CLIP) to score states in relation to given commands during training. These scores are used to re-weight the rewards of the underlying skill discovery algorithm. **Left**: In state-based tasks (top row), task descriptions are provided to foundation models, which are queried to generate a score function $f(s)$ based on our requirements. In pixel-based tasks (bottom row), the current visual state, textual descriptions of desirable and undesirable intentions are input to foundation models to obtain embeddings. These embeddings are then used to form the score function $f(s)$, see Equation (8). **Right**: During training, skill discovery rewards are re-weighted using the score function. Re-weighting rewards of the underlying skill discovery method (we use METRA (Park et al., 2023b)) by the score function is equivalent with using the score function as the distance metric in the DSD objective.

els cannot handle visual input) and requires step-wise chat-style querying during training, which is extremely expensive.

To address these limitations, in this work, we leverage the recent success on foundation models (Radford et al., 2021; Ouyang et al., 2022) and introduce a **Fo**undation model **G**uided (FoG) skill discovery method (see Figure 1). More specifically, we propose to extract a score function from foundation models (in an one-time or batch-forwarding manner) to score states based on our intentions, assigning higher scores for desirable behaviors and lower for undesirable ones. These scores are then used to re-weight rewards of unsupervised skill discovery algorithms. By optimizing the re-weighted rewards, FoG learns not only to maximize skill diversity, but also to satisfy given human intentions. Our results show that FoG successfully learns to follow human guidance on a variety of state-based and pixel-based tasks, including high-dimensional humanoid control.

Our main contributions are threefold: 1) We introduce a novel foundation model guided unsupervised skill discovery method (FoG), which leverages foundation models to incorporate human intentions into skill discovery. Unlike most previous methods that focus solely on maximizing skill diversity, FoG not only learns diverse skills but also aligns them with specified human intentions. 2) We evaluate FoG alongside three state-of-the-art baselines on both state-based and pixel-based tasks. FoG outperforms baselines in both scenarios, showcasing superior generalization. 3) We show FoG can learn behaviors that are challenging to define, such as being 'twisted' and 'stretched' on a humanoid robot, suggesting its potential for more complex applications. The FoG codebase can be found in the supplemental materials.

## 2 PRELIMINARIES AND PROBLEM SETTING

We consider a reward-free Markov Decision Process defined as $\mathcal{M} = (\mathcal{A}, \mathcal{S}, p)$. $\mathcal{S}$ denotes the state space, $\mathcal{A}$ denotes the action space and $p$ is the transition dynamics function. A latent vector $z \in \mathcal{Z}$ (also called 'skill') is sampled during training and its conditioned policy $\pi(\cdot|s, z)$ is executed to get a skill trajectory $\tau = (s_0, s_1, ..., s_T)$ following the process: $p(\tau|z) = p(s_0) \prod_{t=0}^{T-1} \pi(a_t|s_t, z)p(s_{t+1}|s_t)$. $\pi(\cdot|s, z)$ can be learned by optimizing unsupervised exploration objectives we discussed in Section 5, based on mutual information or distance-maximization.

FoG utilizes the Distance-maximizing Skill Discovery (DSD) (Park et al., 2023a) objective. Unlike mutual information based methods (Eysenbach et al., 2018), DSD aims to maximize the Wasserstein dependency measure (WDM) (Ozair et al., 2019) defined as:

$$I_{\mathcal{W}}(S; Z) = \mathcal{W}(p(s, z), p(s)p(z)), \tag{1}$$

where $\mathcal{W}$ is the 1-Wasserstein distance on the metric space $(S \times Z, d)$ for distance metric $d$. By maximizing the objective in Equation (1), the agent will not only maximize the diversity of skills, but also maximize the distance metric $d$. Under some simplifying assumptions (Ozair et al., 2019; Villani et al., 2009), maximization of Equation (1) can then be rewritten as:

$$\sup_{\pi,\phi} \mathbb{E}_{p(\tau,z)} \left[ \sum_{t=0}^{T-1} \left( \phi(s') - \phi(s) \right)^{\top} z \right] \quad \text{s.t.} \quad \|\phi(x) - \phi(y)\|_2 \le d(x,y), \quad \forall(x,y) \in S, \quad (2)$$

where $\phi$ is a function that maps states to a $D$-dimensional space, which is the same as the skill space $Z$. Intuitively, Equation (2) aims to align the direction of $z$ and $\phi(s') - \phi(s)$ (to learn distinguishable and diverse skills), while maximizing the length of $\|\phi(s') - \phi(s)\|$, which leads to an increase in the distance between states based on the given distance metric $d$ due to the Lipschitz constraint (Park et al., 2023a). In principle, $d(x,y)$ in Equation (2) can be replaced by any of the distance metrics in Table 1, resulting in different unsupervised skill discovery methods. Equation (2) can be optimized with dual gradient descent, incorporating a Lagrange multiplier $\lambda$ and a small slack variable $\epsilon > 0$:

$$\text{Update } \phi \text{ to maximize:} \qquad \mathbb{E}[(\phi(s') - \phi(s))^{\top} z] + \lambda \cdot \min(\epsilon, d(s,s') - \|\phi(s) - \phi(s')\|) \quad (3)$$

$$\text{Update } \lambda \text{ to minimize:} \qquad -\lambda \cdot \mathbb{E}[\min(\epsilon, d(s,s') - \|\phi(s) - \phi(s')\|)] \quad (4)$$

$$\text{Update } \pi \text{ with reward:} \qquad (\phi(s') - \phi(s))^{\top} z \quad (5)$$

For derivation of these equations we refer to Park et al. (2022; 2023a;b).

## 3 Foundation Model Guided Skill Discovery

The key idea of FoG is to extract a score function from foundation models based on human intentions to re-weight skill discovery rewards. This process is illustrated in Figure 1. In state-based tasks, the foundation model is queried to output a score function that meets our intentions. In pixel-based tasks, state embedding and human intentional text embedding of a foundation models are used to form the score function. During unsupervised skill discovery, the skill-conditioned policy is trained to maximize the re-weighted rewards of the underlying skill discovery algorithm.

### 3.1 Score Function

We extract a score function from foundation models that can assign higher values for desirable states and lower values for undesirable states with respect to the given intentions. This score function is then used to reweight rewards of the underlying skill discovery method. By optimizing the reweighted rewards, agents will learn skills that are both diverse and desirable. We define the score function $f : S \to [0,1]$ which takes a state as input and outputs a value between 0 and 1, indicating the desirability of the given state. This score function is then used to reweight the skill discovery rewards. The skill discovery reward $r_{skill}$ of Equation (5) therefore becomes:

$$r = f(s') \times r_{skill} = f(s')(\phi(s') - \phi(s))^{\top} z, \quad (6)$$

where we care about the states $s'$ the agent reaches instead of the state $s$ the agent comes from. Thus, the score function $f$ takes $s'$ as the input. Since we use METRA (Park et al., 2023b) as the underlying skill discovery algorithm, and use the score function to re-weight the METRA rewards, this is equivalent to using it as the distance metric in the DSD objective:

$$\sup_{\pi,\phi} \mathbb{E}_{p(\tau,z)} \left[ \sum_{t=0}^{T-1} \left( \phi(s_{t+1}) - \phi(s_t) \right)^{\top} z \right] \quad \text{s.t.} \quad \|\phi(s) - \phi(s')\|_2 \le f(s'), \quad \forall(s,s') \in S_{adj}, \quad (7)$$

where $S_{adj}$ represents the set of adjacent state pairs. The derivation of Equation (7) can be found in Appendix A.1. By using the score function as the distance metric in the DSD objective, FoG not only maximizes the diversity of skills, but also maximizes the output of the score function, leading to skills that are more aligned with our intentions.

In practice, we find that a binary score function works well, i.e. outputting 1 if the state is desirable and $\alpha$ if it is not, where $0 \le \alpha < 1$. We examine different values of $\alpha$ and a non-binary score function in Section 4.2.4.

## 3.2 IMPLEMENTATION DETAILS

Our work builds on top of METRA (Park et al., 2023b), which is the state-of-the-art unsupervised skill discovery method that works for both state-based and pixel-based input. FoG re-weights the skill discovery reward of METRA by the score function that is extracted from foundation models. For state-based tasks, we ask foundation models to generate the score function directly. For pixel-based tasks, we use foundation models to output embeddings to form the score function. All code is available through the supplemental materials.

**State-based:** We ask ChatGPT or Claude to generate a score function $f(s)$ that equals 1 if the state satisfies our intentions, and $\alpha$ otherwise. Unlike Eureka (Ma et al., 2023), which queries foundation models to generate a reward function for training agents from scratch, FoG instead asks for a score function to modulate skill discovery. Prompt details for state-based tasks and examples of resulting output score functions are provided in Appendix A.12.

**Pixel-based:** We use CLIP (Radford et al., 2021), a vision-language model that is trained to align images and text, to first generate embedding for images (pixel-based states) and texts (textual descriptions of our intentions). Then, the score function is formed by computing the $Cosine$ similarity between the image and text embedding. If the current state is more similar to the description of the desirable intention, the output is 1. Conversely, if it is more similar to the undesirable one, the output is $\alpha$. The score function can be expressed as Equation (8).

$$f(s) = \begin{cases} 1, & \text{if } Cosine(E_s, E_{text1}) > Cosine(E_s, E_{text2}). \\ \alpha, & \text{otherwise.} \end{cases} \tag{8}$$

where $E_s$ is the embedding of the current pixel-based state, $E_{text1}$ and $E_{text2}$ are the embedding of the textual descriptions of desirable and undesirable intentions, respectively. Setting $\alpha = 0$ attempts to not learn undesirable behaviors at all (since $\alpha \times r_{skill} = 0$) while setting $\alpha = 1$ reduces FoG to the underlying skill discovery algorithm METRA. We examine different values of $\alpha$ in Section 4.2. Details of textual descriptions of desirable and undesirable intentions can be found in Appendix A.8.

## 4 EXPERIMENTS

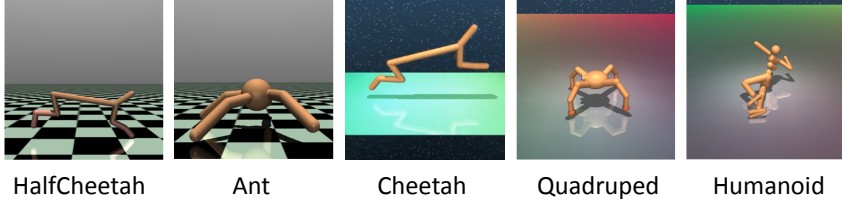

|  |  |  |  |  |
|---|---|---|---|---|
| HalfCheetah | Ant | Cheetah | Quadruped | Humanoid |

Figure 2: Environments used in our work. HalfCheetah and Ant are state-based, and other three are pixel-based.

Through our experiments, we aim to answer the following questions: 1) How does FoG perform in state-based tasks where more context and informative features are provided? 2) In pixel-based tasks, where only visual information is provided, can FoG guide agents to learn diverse and desirable behaviors and skills?

We use environments that are commonly used in unsupervised skill discovery literature, see Figure 2, including two state-based tasks and three pixel-based tasks: HalfCheetah and Ant are state-based tasks from OpenAI gym (Brockman et al., 2016), Cheetah, Quadruped and Humanoid are pixel-based tasks from DMC (Tunyasuvunakool et al., 2020).

FoG falls into the category of unsupervised skill discovery methods, but with human intentions integrated. DoDont (Kim et al., 2024b) and LGSD (Rho et al., 2024) are the most relevant baselines, as both incorporate human intentions to skill discovery in different ways. DoDont relies on demonstration videos, which can be difficult to obtain in certain scenarios. LGSD uses large language models, which needs step-wise chat-style query and are limited to state-based tasks. For these reasons, we skip LGSD and use DoDont as one of the baselines instead. Overall, we have five baselines for FoG to compare against: 1) METRA (Park et al., 2023b), the state-of-the-art unsupervised skill discovery

method; 2) METRA+, which uses hand-defined reward functions as score functions, and was also used as a baseline in DoDont (Kim et al., 2024b); 3) LSD (Park et al., 2022), an unsupervised skill discovery method that maximizes DSD objective with Euclidean distance as the distance metric; 4) DoDont (Kim et al., 2024b), a demonstration-guided unsupervised skill discovery method, learns diverse and desirable behaviors shown in the demonstrations; 5) DoDont+, a variant of DoDont that replaces expert demonstrations with demonstrations annotated using foundation models.

All agents in the same task are trained with the same number of environment steps and all experiments are performed three times with different independent seeds, and average results with error bars are reported. For simplicity, we set $\alpha = 0$ for all experiments. More details of environments and baseline implementations can be found in Appendix A.9. See website[1] for videos of the learned behaviors and skills.

## 4.1 STATE-BASED TASKS

To test whether FoG can work in state-based tasks, we train FoG in HalfCheetah and Ant. Following the details in Section 3.2, we input the description of the tasks, information about state space and action space to foundation models as context, then ask foundation models to generate a score function that returns 1 when the requirement in the query is satisfied otherwise $\alpha$. In HalfCheetah, we train FoG to eliminate dangerous behaviors (flipping over). In Ant, we train FoG to avoid a specific area, in this case to not go south.

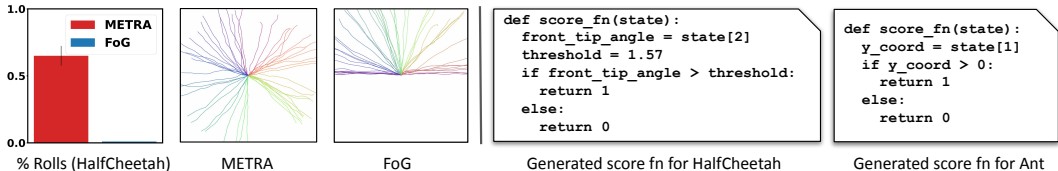

Figure 3: **Left**: Comparison between METRA and FoG on two state-based tasks, HalfCheetah and Ant. FoG learns not to roll in HalfCheetah and not move to south in Ant, while METRA rolls more than $50\%$ of the time in HalfCheetah and goes to all directions, violating our intention in Ant. **Right**: Score functions generated by foundation models which are uesed to re-weight skill discovery rewards. In both HalfCheetah and Ant, foundation models successfully capture the relevant state dimension and set threshold for it.

Results of these experiments are visualized in Figure 3, with generated score functions for both tasks at the right. We first of all see that foundation models can recognize feature dimensions of the state that are important for meeting our requirements. For example, in HalfCheetah the second dimension of the state is the angle of Cheetah's front tip, which is important for determining if the agent flips over or not. In Ant, the first dimension of the state is the y-coordinate of Ant, which can be used to locate the agent in a south-north position. We see foundation models clearly set the right threshold and implement the logic to fulfil the intention we asked for, i.e., if the angle of the Cheetah's front tip is larger than $1.57$ in radians (90 degrees) it flips over, and if the y-coordinate of Ant is larger than 0 it is in the north part of the plane. By re-weighting the skill discovery rewards using the generated score function from foundation models, FoG learns to not roll in Cheetah while METRA flips a lot (left graph of Figure 3). In Ant, FoG learns to always move to north and METRA learns to go every directions (second and third graph in Figure 3).

## 4.2 PIXEL-BASED TASKS

We now conduct experiments in pixel-based tasks, where only visual information is available. Unlike in state-based tasks, where we ask foundation models to directly generate a score function, in pixel-based tasks we leverage foundation models to output embeddings of 1) the visual state and 2) textual descriptions of our desirable and undesirable intentions. The score function is then computed from Equation (8). We examine FoG in three aspects: a) Can it learn to eliminate undesirable behaviors? b) Can it learn to avoid certain areas? c) Can it learn complex behaviors that are difficult to clearly define?

---

[1]https://sites.google.com/view/iclr-fog

### 4.2.1 LEARN TO ELIMINATE UNDESIRABLE BEHAVIORS

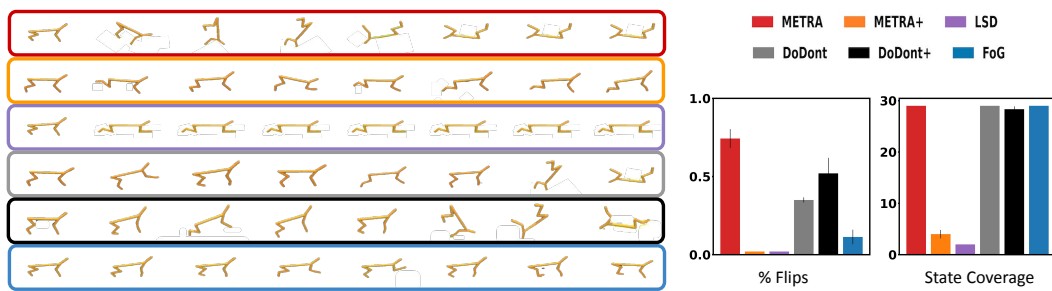

Figure 4: **Left**: Executions of example skills from different agents in pixel-based environment, Cheetah. From top to bottom: METRA, METRA+, LSD, DoDont, DoDont+, FoG. **Right**: Percentage of flips (which should be prevented based on the guidance) and state coverage for different agents. METRA, METRA+, DoDont and DoDont+ learn to discover diverse states, but suffer from frequent flipping. LSD fails to learn diverse skills. FoG learns to run without flipping, achieving both high state coverage and low flipping percentage.

We first focus on guiding the agent to learn desirable low-level behaviors (e.g., standing normal) while eliminating undesirable ones (e.g., flipping over) that could potentially damage the robot. In pixel-based Cheetah, we use 'agent flips over' and 'agent stands normally' as textual descriptions to express our intentions. The score function is then formed according to Equation (8).

As shown in the left figure of Figure 4, FoG (bottom) consistently learns to run without flipping, demonstrating the lowest percentage of flips during evaluation. In contrast, other methods struggle to prevent flipping effectively. METRA flips in over 70% of episodes, DoDont in more than 35%, and DoDont+ in 50% of the episodes. LSD and METRA+ struggle to learn to move in different directions, discovering static behaviors and rarely flipping. Although METRA, DoDont, DoDont+ and FoG achieve similar state coverage, FoG effectively prevents flipping.

The poor performance of METRA+ suggests that defining a proper score function manually is not trivial (we follow the definition in (Kim et al., 2024b) and use $r_{run} - r_{flip}$ as the score function). The poor performance of DoDont stems from the inaccurate classifier, which exploits the color of the ground to distinguish different states (normal and flipping postures), outputting high scores for unseen undesirable behaviors. To evaluate how these learned skills perform in downstream tasks, we train a controller to select from the learned set of skills. This controller trained using FoG skills shows quick adaptations in the downstream tasks, as shown in Appendix A.3.

### 4.2.2 LEARN TO AVOID HAZARDOUS AREAS

Previous unsupervised skill discovery methods focus solely on maximizing skill diversity, often leading agents to explore to all possible directions. In practice, however, we want agents to avoid certain areas when they are hazardous. For instance, a robot operating in a factory should be able to avoid prohibited areas. To test whether FoG can learn to avoid certain areas (high-level policies, as opposed to low-level behaviors in Section 4.2.1), we train FoG in the pixel-based versions of Cheetah and Quadruped. We designate the right area in Cheetah and bottom-left area in Quadruped are hazardous and train agents to avoid them. Since there are no explicit indicators of directions in these two tasks, we express our intentions through colors. For example, in Cheetah, we use descriptions like 'ground is blue' and 'ground is orange' to signal whether the agent is on the left or right part and then form the score function following Equation (8).

Figure 5 illustrates the learned skills and 'Safe State Coverage' (the coverage of safe areas minus that of hazardous areas) of different agents. FoG clearly biases movement toward the safe areas. In Cheetah it prefers to go to the left part and in Quadruped it avoids the bottom-left area, resulting in higher safe state coverage than the baselines. However, METRA explores all directions indiscriminately and LSD fails to move, leading to the lowest safe state coverage. DoDont performs well in Quadruped but not in Cheetah (the classifier are unsure about initial states thus harm the exploration). The slightly worse performance of DoDont+ (compared to DoDont) in Quadruped stems

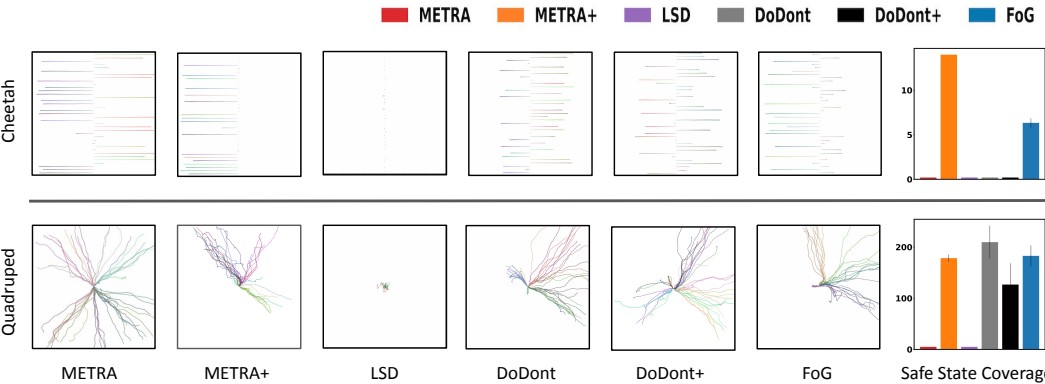

Figure 5: **Top**: Results on the pixel-based environment Cheetah, with learned skills shown in x-coordinates. METRA+ learns to perfectly avoid the undesirable area and FoG has a strong preference to go to the desirable area, as also clearly visible from the Safe State Coverage on the right. Other agents fail. **Bottom**: Results on the pixel-based environment Quadruped, with learned skills shown as xy-coordinates. Similar conclusions can be drawn regarding most of agents. Unlike in Cheetah, DoDont successfully learns to avoid the bottom-left areas.

from its inaccurate demonstrations annotated by foundation models. METRA+ performs the best, likely because that defining a score function in these tasks is straightforward (assigning 1 to states in safe regions and 0 for ones in hazardous regions (Kim et al., 2024b)). The results suggest that with expert-level demonstrations and 'perfect' hand-crafted score function, DoDont and METRA+ could potentially outperform FoG. However, the strength of FoG shines in scenarios where obtaining expert-level demonstrations or crafting a perfect score function is challenging, which is generally the case.

Non-expert demonstrations (like ones annotated by foundation models, which are used in DoDont+) introduce inaccuracies to the classifier, with annotation accuracy around 70%. This leads to an inaccurate classifier that consistently generates unreliable signals, ultimately resulting in poor performance. In contrast, FoG leverages CLIP on-the-fly. Although CLIP does not achieve perfect accuracy, FoG adapts and aligns its learning process with the overall positive trend of CLIP's output. This creates a self-correcting loop, culminating in significantly better performance.

### 4.2.3 LEARN TO TWIST IN HUMANOID

Humanoid is a challenging high-dimensional control task with a 21-D action space. Defining when this humanoid robot is "twisted" or "stretched" is both hard and subjective. This also makes it hard to design a reward function that can guide the agent to learn such behaviors. Therefore, we instead leverage a foundation model to determine when the agent is twisted or stretched, assigning a higher score for the former and a lower score for the latter. As such FoG can discover intricate behaviours that are hard to explicitly define, such as being "twisted" or "stretched". We could not

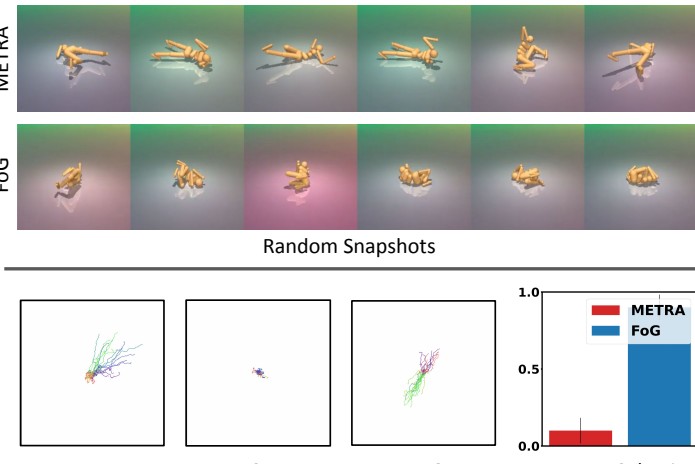

Figure 6: **Top**: Random snapshots during evaluation of METRA and FoG in pixel-based Humanoid. **Bottom**: Learned skills (shown in xy-coordinates) of different agents and results on human participants. Humans pick FoG to be more "twisted" 90% of the time.

compare FoG with DoDont (Kim et al., 2024b) as the original paper does not include results on Humanoid, probably because demonstrations of a humanoid robot are challenging to obtain (an issue we also encountered).

We train FoG in the Humanoid task using intention descriptions 'agent is stretched' and 'agent is twisted', and form the score function according to Equation (8). To quantitatively assess whether the agent has successfully learned to twist, we create a questionnaire and ask ten human participants to evaluate videos of different agents, selecting the ones they perceive as more "twisted". Videos and the questionnaire can be found on the project website and details of the experimental setup can be found in Appendix A.11.

In the top part of Figure 6, it is clear that FoG learns to exhibits more "twisted" postures while ME-TRA tends to appear more "stretched". Both FoG and METRA successfully learn to move in different directions, highlighting the diversity of the learned skills. The 'Human Selection' (right-bottom of Figure 6) shows how participants perceive the trained skills, with 90% of the time participants selecting FoG as more "twisted", further validating the observed outcomes. FoG's ability to move in different directions with "twisted" postures suggests its potential to guide agents in discovering skills involving behaviors with subjective definitions.

### 4.2.4 ABLATION STUDY

FoG introduces two new hyperparameters. The first, $\alpha$ in the binary score function of Equation (8), controls how much the skill discovery rewards are re-weighted when the state is undesirable. Higher values assign greater weights to undesirable states, making rewards for these states less distinguishable with those of desirable states. As a result, agents are more likely to learn undesirable behaviors. We evaluate three values, $\alpha = 0, 0.5, 0.8$. Meanwhile, since we compute similarities between visual images and texts, shown in Equation (8), rather than forming a binary function, we can also softmax both similarities and directly use them to re-weight (see Equation (12)) the skill discovery rewards. The left part of Figure 7 shows flip percentages of FoG agents trained with different $\alpha$ values in the Cheetah task. Unsurprisingly, higher $\alpha$ values lead to more undesirable behaviors: the agent flips more often. Directly using similarity (**sim**) to re-weight skill discovery rewards returns poor performance. Although setting $\alpha = 0$ works well across all experiments performed in this work, in some cases, it could be too strict and weaken exploration. See extra results in Appendix A.5.

When we use FoG in pixel-based tasks, obtaining embeddings for every pixel state is computationally expensive. In practice, we calculate embeddings for every $N$th state and apply the score for that state to the following $(N-1)$ states. Smaller $N$ values result in more accurate scores but increase computational costs. The right part of Figure 7 therefore shows performance for three different values of $N$. Smaller $N$ values score states more frequently, leading to more accurate scores and improved performance (fewer flips). However, there is no significant difference in performance between $N = 10$ and $N = 20$, suggesting behaviors in Cheetah are quite smooth and skipping 10 or 20 states leads to similar results.

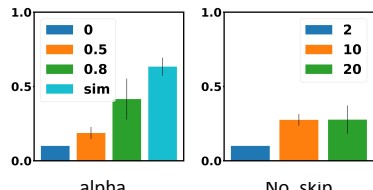

Figure 7: Percentages of flips that different FoG shows on the Cheetah environment. Smaller $\alpha$ and $N$ return better performance.

## 5 RELATED WORK

**Unsupervised Skill discovery**: FoG builds on top of unsupervised skill discovery methods, allowing agents to learn diverse skills without the use of hand-crafted reward functions. One line of research in unsupervised skill discovery focuses on maximizing mutual information $I(\cdot; \cdot)$ between skills $Z$ and states $S$, i.e., $I(S; Z) = H(S) - H(S|Z) = H(Z) - H(Z|S)$, where $H(\cdot)$ denotes entropy. By associating states $s \in S$ with different latent skill vectors $z \in Z$, these methods learns diverse skills that are mutually distinct (Eysenbach et al., 2018; Sharma et al., 2019; Laskin et al., 2022). SASD (Kim et al., 2023) integrates a pre-defined safety-indicator function into the critic learning process of mutual-information based unsupervised skill discovery methods, enabling the learning of safe behaviors. EDL (Hussonnois et al., 2023) employs preference-based RL to integrate human preferences, requiring a human-in-the-loop to continuously provide feedback on trajectory segments. Both methods depend on human effort, either to pre-define the safety-indicator function

Table 1: Distance metrics used by different methods in the distance-maximizing skill discovery objective. $q_\theta$ is a density function parameterized by $\theta$. Temporal distance is defined as the minimum number of environmental steps needed for the agent to go from one state to another state. $s_{lang}$ is the textual description of the state $s$. $p_\varphi$ is a classifier parameterized by $\varphi$.

| LSD | CSD | METRA | LSGD | DoDont | ours |
|---|---|---|---|---|---|
| $\|s' - s\|$ | $-\log q_\theta(s'\|s)$ | temporal distance | $distance(s'_{lang}, s_{lang})$ | $p_\varphi(s', s)$ | score fn |

or to actively provide preferences, and they operate only with state-based input. In contrast, FoG eliminates the need for human involvement and supports both state and pixel-based input.

Aforementioned mutual information-based methods do not always encourage the agent to discover distant states, as the mutual information objective can be satisfied by learning simple and static skills (Park et al., 2023b; 2022). To address this limitation, Park et al. (2023a) introduces a Distance-maximizing Skill Discovery (DSD) framework that learns diverse skills while maximizing the traveled distance under the given distance metric $d$. LSD (Park et al., 2022) uses Euclidean distance between states as the distance metric to encourage agents to visit states that are as far apart as possible. CSD (Park et al., 2023a) employs a density function over visited states as the distance metric, to encourage agents to visit less frequently visited states. However, LSD and CSD only work with state-based inputs and fail in pixel-based tasks. METRA (Park et al., 2023b) instead uses a temporal distance function that is applicable in visual tasks as well, as the distance metric to push the agent to discover states that are temporally far apart. LGSD (Rho et al., 2024) utilizes foundation models to first convert state-based inputs to text descriptions, then uses embedding distance between text descriptions as the distance metric to encourage agents to learn semantic diverse skills. DoDont (Kim et al., 2024b) employs demonstrations to guide agents in learning desirable behaviors. Specifically, it trains a classifier over the demonstrations of what the agent should and should not do, and uses it as a distance metric in DSD, encouraging agents to learn to maximize intentions of the given demonstrations. Some distance metrics used by different methods are summarized in Table 1. Note that FoG can be interpreted as using a score function extracted from foundation models as the distance metric in DSD. We refer to Section 3 for further details.

FoG is most closely related to DoDont and LGSD, as both these methods aim to incorporate human preferences into skill discovery. However, DoDont requires expert demonstrations, which can be expensive to obtain, and it only works well with state-based inputs when incorporating behavioral intentions. LGSD uses language models so only works for state-based tasks (language models cannot handle visual input), and querying large language models in a step-wise chat-style manner is expensive. In contrast, FoG leverages vision-language models and extracts a score function from them in a one-time (in state-based tasks) or batch-forwarding manner (in pixel-based tasks) to reweight the underlying skill discovery rewards. It therefore has a fast response time and works well in both state-based and pixel-based tasks.

**Foundation Models in Reinforcement Learning**: FoG leverages foundation models to guide unsupervised skill discovery in learning desirable behaviors. Thanks to success of foundation models (Touvron et al., 2023; Liu et al., 2023b) they can now be used to provide information for RL agents. Motif (Klissarov et al., 2023) employs large language models to generate exploration bonuses in the text-based game NetHack (Küttler et al., 2020). Eureka (Ma et al., 2023) uses large language models to generate reward functions for state-based robotic tasks, outperforming human designed reward functions across multiple tasks. IGE (Lu et al., 2024) leverages large language models to select interesting goals and propose actions in state-based environments with text descriptions. LAMP (Adeniji et al., 2023) utilizes the similarity between pixel embedding and text-commands embedding, as output by a vision-language model, as the reward to pre-train agents in visual robotic tasks. RoboCLIP (Sontakke et al., 2024) uses a video-language model to label exploration trajectories as either success or failure, supplying sparse rewards in visual robotic tasks. Additionally, Rocamonde et al. (2023) investigate how vision-language models can be used as reward models in a zero-shot manner. These approaches utilize existing pre-trained foundation models in a zero-shot manner, without fine-tuning them for specific tasks. However, to achieve better performance, other works such as Minedojo (Fan et al., 2022) and EmbodiedGPT (Mu et al., 2024) train foundation models from scratch or fine-tune them on specific downstream tasks (Adeniji et al., 2023). Despite this, FoG demonstrates that pre-trained foundation models, even without fine-tuning, can be used to

guide RL agents to discover diverse and desirable skills. FoG is evaluated in both state-based and pixel-based tasks. For state-based tasks, we ask foundation models to generate a score function that meets our intentions. FoG therefore differs from, e.g., Eureka (Ma et al., 2023) in two key ways: 1) FoG does not require iterative feedback from the environment. Eureka involves providing feedback multiple times and adjusting the reward function iteratively. 2) FoG leverages foundation models to generate a score function, which is used to re-weight underlying skill discovery rewards. In contrast, Eureka generates a reward function that is directly used for training agents.

## 6 Limitations and Future Work

Although FoG performs well, it is not without limitations. FoG employs foundation models to guide desirable skill discovery. However, there is no $100\%$ guarantee that score functions generated by foundation models are always appropriate. Additionally, since the score function is defined based on individual states, FoG may struggle to capture process-based alignment. This limitation could be addressed by defining the score function over a sequence of states. For example, RoboCLIP (Sontakke et al., 2024) utilizes foundation models to provide sparse reward signals based on performance of the whole episode.

Furthermore, FoG uses CLIP (Radford et al., 2021), a well-established vision-language model for pixel-based tasks. We believe FoG could benefit from more advanced foundation models (Liu et al., 2023a; Yao et al., 2024) or task-specific models (Padalkar et al., 2023; Valevski et al., 2024). One could also explore the performance of FoG with more complex intentions. In addition, FoG should also be scaled to more challenging tasks. Some preliminary results can be found in Appendices A.6 and A.7.

## 7 Conclusion

We propose a novel unsupervised skill discovery method, FoG, guided by foundation models to incorporate human intentions. FoG first extracts a score function from foundation models based on input intentions, assigning higher preference to desirable states and lower preference to undesirable ones. This score function is then used to re-weight the underlying skill discovery rewards. By optimizing re-weighted rewards, FoG discovers not only diverse but also desirable skills. In addition, we also show FoG can learn skills involving behaviors that are complex and subjectively defined. We hope FoG inspires future efforts in incorporating human intentions in unsupervised skill discovery.

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

## A  APPENDIX

### A.1  DERIVATION OF EQUATION (7)

The original DSD objective is shown in Equation (2). It is crucial to define a appropriate distance metric to encourage agents to not only learn diverse skills but also maximize the given distance metric. Park et al. (2023b) uses the temporal distance as the distance metric for the DSD objective in METRA, shown in Equation (9).

$$\sup_{\pi,\phi} \mathbb{E}_{p(\tau,z)} \left[ \sum_{t=0}^{T-1} \left(\phi(s_{t+1}) - \phi(s_t)\right)^\top z \right] \quad \text{s.t.} \ \ \|\phi(s) - \phi(s')\|_2 \leq 1, \ \ \forall (s,s') \in S_{adj}. \quad (9)$$

Now, we use the score function $f(s')$ to re-weight the METRA rewards to get the objective of FoG. The new objective (FoG) now becomes Equation (10):

$$\sup_{\pi,\phi} \mathbb{E}_{p(\tau,z)} \left[ \sum_{t=0}^{T-1} f(s') \left(\phi(s_{t+1}) - \phi(s_t)\right)^\top z \right] \quad \text{s.t.} \ \ \|\phi(s) - \phi(s')\|_2 \leq 1, \ \ \forall (s,s') \in S_{adj}. \quad (10)$$

Following Kim et al. (2024a), let scaled state function $\tilde{\phi}(s) = \phi(s)f(s)$. By replacing $\phi(s)$ with $\tilde{\phi}(s)/f(s)$ and transforming the constraint in Equation (10) (since $f(s) \geq 0$), we derive Equation (11) (Equation (7)), which is using the score function as the distance metric in the DSD objective.

$$\sup_{\pi,\phi} \mathbb{E}_{p(\tau,z)} \left[ \sum_{t=0}^{T-1} \left(\tilde{\phi}(s_{t+1}) - \tilde{\phi}(s_t)\right)^\top z \right] \quad \text{s.t.} \ \ \|\tilde{\phi}(s) - \tilde{\phi}(s')\|_2 \leq f(s'), \ \ \forall (s,s') \in S_{adj}. \quad (11)$$

Hereby, we show that using the score function to re-weight the METRA rewards is equivalent as using it as the distance metric in the DSD objective.

### A.2  NON-BINARY SCORE FUNCTION

Instead of using a binary score function in Equation (8), we can also form a non-binary score function.

$$f(s) = \frac{e^{Cosine(E_s, E_{text1})}}{e^{Cosine(E_s, E_{text1})} + e^{Cosine(E_s, E_{text2})}}, \quad (12)$$

where $E_s$ is the embedding of the current pixel-based state, $E_{text1}$ is the embedding of textual descriptions of the desirable intention and $E_{text2}$ is the embedding of textual descriptions of the undesirable intention.

### A.3  DOWNSTREAM TASKS

After obtaining skills, we can train a controller to select these (frozen) learned skills to achieve given downstream goals. We follow the implementation of Park et al. (2023b), and set $g \sim [-10, 10]$ as the goal. During training, the agent receives a reward of 10 if the goal is reached. We train a controller to select a skill $z$ every $K = 50$ steps, and the learned policy $\pi(\cdot|s,z)$ is executed for $K$ steps. We use SAC (Haarnoja et al., 2018) for training the controller and all hyperparameters are kept the same as the METRA codebase. Results are shown in Figure 8. The controller that is trained using frozen skills learned by FoG shows better performance at the beginning and converges faster than the baselines, indicating that FoG effectively learns meaningful skills that can be quickly adapted to downstream tasks. LSD does not learn useful skills thus the trained controller performs poorly. METRA slightly lags behind of DoDont.

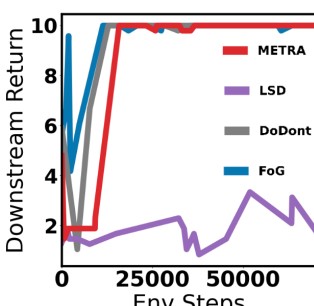

Figure 8: Downstream task performance.

### A.4 FOUNDATION MODELS

For state-based tasks, we query ChatGPT[2] or Cluade[3] to generate score functions that meet our requirements. For pixel-based tasks, we use pre-trained CLIP (clip-vit-large-patch14) from huggingface[4].

### A.5 QUADRUPED LEARNS TO NOT FLIP

Although we found that setting $\alpha = 0$ works well in experiments presented in Section 4, sometimes it might hurt the exploration. Similar with experiments performed in Section 4.2.1, here, we train FoG to not flip in Quadruped. We see in Figure 9, FoG learns to not flip most of time (less than $20\%$) when setting $\alpha = 0$, but it almost always stays near the starting point and does not explore, resulting in low state coverage. After loosing $\alpha$ a bit and set it to $0.1$, FoG learns to eliminate all flips and has a significant higher state coverage.

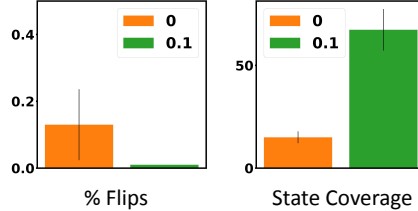

Figure 9: Results on the Quadruped task. Setting $\alpha = 0$ explores less (lower state coverage) thus results in worse performance (more flips).

### A.6 RESULTS ON FRANKA KITCHEN

To examine FoG in more complicated tasks, we train FoG in Franka Kitchen (introduced by Gupta et al. (2019)) with different textual descriptions of intentions, such as `robotic arm is stretched`, `robotic arm is twisted` and `robotic arm is on the right of the scene`. Results can be seen in Figure 10. By using different intentions, we see robotic arms clearly bias the movements to different areas. However, we did not find a way to use these skills to better solve the downstream tasks yet. We hope this could inspire future efforts in investigating FoG in more complex tasks.

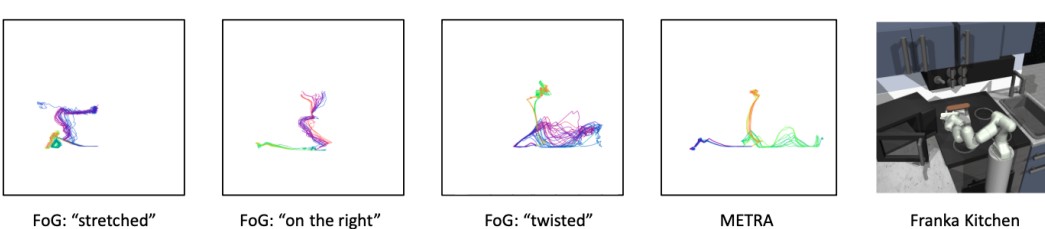

FoG: "stretched"     FoG: "on the right"     FoG: "twisted"     METRA     Franka Kitchen

Figure 10: In Franka Kitchen, different skills FoG learned with different textual descriptions of intentions. Skills are displayed with x-y coordinates of the robotic arm.

### A.7 RESULTS ON MULTIPLE INTENTIONS

In Section 4, only one intention is used in FoG. In principle, multiple intentions could be used simultaneously to form the score function. Then, Equation (8) becomes:

$$f(s) = \begin{cases} 1, & \text{if } Cosine(E_s, E_{text1}^1) > Cosine(E_s, E_{text2}^1) \text{ and} \\ & \quad Cosine(E_s, E_{text1}^2) > Cosine(E_s, E_{text2}^2) \text{ and} \\ & \quad ... \\ & \quad Cosine(E_s, E_{text1}^n) > Cosine(E_s, E_{text2}^n) \\ \alpha, & \text{otherwise.} \end{cases} \tag{13}$$

where $E_{text1}^n$ and $E_{text2}^n$ are the $n$th textual descriptions of our intentions. Now, the score function $f(s)$ only assigns higher values to desirable states when all provided intentions are satisfied. For

---

[2]https://chatgpt.com

[3]https://claude.ai/new

[4]https://huggingface.co/openai/clip-vit-large-patch14

example, we could ask FoG to not only learns to not flip, but also to avoid the right area. The textual descriptions we should use are: 1) `agent flips over`, `agent stands normally`; 2) `ground is Yellow-Orange`, `ground is Green-Blue`. See the result in Figure 11, the agent does not learn to avoid the right part at all but it does learn to eliminate flips (not shown in the figure). We found that using multiple intentions restricts the exploration too much so that the agent might just learn to fulfill one intention and ignore others or ignore all of them and learns to not move at all. Using multiple intentions in FoG still needs more investigations and we hope the preliminary results and ideas presented in this section could inspire future efforts.

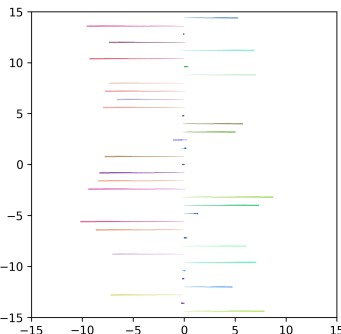

Figure 11: Skills learned by FoG with two intentions, i.e. 1) not flip; 2) not go right.

### A.8 INTENTION PROMPTS USED FOR PIXEL-BASED TASKS

Textual descriptions of intentions we used for Cheetah:

- Section 4.2.1: `The simulated two-leg robot flips over`, `The simulated two-leg robot stands normally`

- Section 4.2.2: `The underneath plane is Yellow-Orange`, `The underneath plane is Green-Blue`

Textual descriptions of intentions we used for Quadruped in Section 4.2.2: `The underneath plane is Pink-Purple`, `The underneath plane is Green-Blue`.

Textual descriptions of intentions we used for Humanoid in Section 4.2.3: `The simulated humanoid robot is stretched`, `The simulated humanoid robot is twisted`.

### A.9 EXPERIMENT DETAILS

#### A.9.1 ENVIRONMENT DETAILS

**State-based:** HalfCheetah and Ant are from OpenAI Gym (Brockman et al., 2016). The state space of HalfCheetah is 18-dimensional and the one of Ant is 29-dimensional. HalfCheetah has a 6-dimensional action space while Ant has a 8-dimensional action space.

**Pixel-based**: Cheetah, Quadruped and Humanoid are from DeepMind Control Suite (Tunyasuvunakool et al., 2020). Following previous work (Lee et al., 2021; Park et al., 2024; 2023b), pixel-based DMC tasks are all with gradient-colored floors to indicate different directions. The size of visual observations is $64 \times 64 \times 3$. The dimension of action space for Cheetah, Quadruped and Humanoid are 6, 12 and 21, respectively. The episode length is 200 for Ant, HalfCheetah and Cheetah, 400 for Quadruped and Humanoid.

### A.9.2 BASELINE DETAILS

**METRA**: We take the official codebase[5] from Park et al. (2023b) and use default hyperparameters for all experiments performed in this paper.

**METRA+**: We follow the implementation of METRA+ in the DoDont paper. For experiments in Section 4.2.1, we use $r_{run} - r_{flip}$ as the reward. For experiments in Section 4.2.2, we assign +1 for the safe region and 0 for the hazardous region.

**LSD**: We take the codebase of METRA, by setting correct arguments (turning off the dual regularization and turning on the spectral normalization), to run LSD. Detailed instructions can be found in the METRA codebase.

**DoDont**: We take the official codebase from Kim et al. (2024b) and implement the training of the instruction net ourselves. We use eight demonstrations for each task, so four for "dos" and four for "donts". Demonstrations are obtained from trained FoG agents and can be found on our project website: `https://sites.google.com/view/iclr-fog`. We stop the training of the classifier after it has more than 97% of accuracy.

**DoDont+**: We use CLIP to score frames (follow Equation (8)) in demonstrations that are used to train DoDont, and assign frames with score of 0 in the "dos" demonstration to "donts" demonstrations, and vice versa. Since CLIP cannot perfectly score frames, some states from "dos" demonstration are moved to "donts" demonstrations, and some states from "donts" demonstration are moved to "dos" demonstration. After training, the classifier has about 70% of accuracy.

### A.9.3 HYPERPARAMETERS DETAILS

We use $\alpha = 0$ and $N = 2$ for all our experiments, unless otherwise mentioned. We train all agents in the same task with the same number of epochs and the performance at the end of training is reported. Details can be seen in Table 2. The same number of episodes is executed in each epoch, and within each episode the same number of environment steps is taken. We train continuous skills and the number of dimensions we used to train all agents in each task can be found in Table 3. We refer readers to read Park et al. (2023b) for details of all used hyperparameters.

| HalfCheetah | Ant | Cheetah | Quadruped | Humanoid |
|---|---|---|---|---|
| 9000 | 9000 | 2000 | 3000 | 4000 |

Table 2: Number of epochs we used for training agents in different environments.

| HalfCheetah | Ant | Cheetah | Quadruped | Humanoid |
|---|---|---|---|---|
| 4 | 2 | 4 | 4 | 2 |

Table 3: Dimensions of skills we used for training skills in different environments.

### A.10 COMPUTATION USAGE

We run our experiments on an internal cluster consisting of A100 and H100 GPUs. Each run takes no more than 24 hours.

### A.11 EXPERIMENTAL SETUP FOR HUMAN JUDGE

In Section 4.2, we train FoG to be twisted in DMC Humanoid task. However, it is difficult for human to design a reward function to measure if learned skills contains more twisted postures or not. Thus, we ask human to be the judge to tell if FoG learns more twisted skills than the ones learned by the baselines.

We pick ten skills of each method randomly, in this case, FoG and METRA, and then pair them randomly. Participates are asked to select the video that shows the most 'twisted' behaviors without

---

[5] `https://github.com/seohongpark/METRA`

given any other information. Please see the full questionnaire we used in `https://sites.google.com/view/iclr-fog/questionnaire-of-humanoid`.

## A.12 Prompts Used for State-based Tasks

**Input to foundation models for HalfCheetah**: [Descriptions of the task, state space and action space] According to the given info, could you please write a python function to check if the cheetah is flipped over or not. If yes, output 1 otherwise 0.

**Possible output**:

```python
import numpy as np

def is_cheetah_flipped(observation):
    """
    Check if the HalfCheetah is flipped over.

    Args:
    observation (np.array): Observation array from the environment.

    Returns:
    int: 1 if the cheetah is flipped over, 0 otherwise.
    """
    # The second element in the observation array is the angle of the front tip
    angle = observation[1]

    # Define the threshold for flipping (e.g., |angle| > π/2)
    threshold = np.pi / 2

    if abs(angle) > threshold:
        return 1
    else:
        return 0

# Example usage:
# Assume `obs` is the current observation from the environment
# obs = env.reset()
# print(is_cheetah_flipped(obs))
```

**Input to foundation models for Ant**: [Descriptions of the task, state space and action space] The agent always starts at [0, 0]. According to the given info, could you please write a python function to check if the agent is in the north part or not. If yes, output 1 otherwise 0.

**Possible Ouput**:

```python
def is_in_north(state):
    """
    Check if the agent is in the northern part (y > 0).

    Parameters:
    state (list or numpy array): The state space observation vector. The y-posit

    Returns:
    int: 1 if the agent is in the northern part, 0 otherwise.
    """
    # Assuming the second element in the state space corresponds to the y-coordi
    y_position = state[1]

    # Check if the y-coordinate is greater than 0
    if y_position > 0:
        return 1
    else:
        return 0
```

