# OpenReview forum: "Guiding Skill Discovery with Foundation Models"
_ICLR.cc/2025/Conference — Submitted to ICLR 2025_

### Official Review · Reviewer_KNUe · 2024-11-01

**Soundness:** 3
**Presentation:** 3
**Contribution:** 2
**Rating:** 3
**Confidence:** 5

**Summary:**

This paper proposes FoG (Foundation model Guided), a skill discovery method in reinforcement learning that incorporates human intentions by leveraging foundation models. FoG uses a score function extracted from foundation models to re-weight skill discovery rewards. This enables learning not only diverse skills but also desirable behaviors. The authors demonstrate FoG's performance compared to existing methods by evaluating it on both state-based and pixel-based tasks. Notably, FoG shows the ability to learn complex behaviors that are difficult to define, such as 'twisted' postures.

**Strengths:**

Unlike existing skill discovery methods, FoG can incorporate human intentions. It demonstrates flexibility by being applicable to both state-based and pixel-based tasks. Furthermore, FoG shows the ability to learn complex behaviors that are difficult to define, such as 'twisted' postures. Through experiments, FoG has been validated to learn intended behaviors, including eliminating dangerous actions and avoiding specific areas. The results demonstrate that FoG successfully guides agents to discover diverse and desirable skills while aligning with specified human intentions.

**Weaknesses:**

* Studies such as [1] have already researched generating reward functions using foundation models. While there is some differentiation in applying a similar idea to skill discovery, the novelty appears limited compared to previous research that generated entire rewards through foundation models. The proposed idea only utilizes foundation models in the restricted aspect of 'applying weights to rewards proposed in other studies'.
* It seems that skills that cannot be generated by METRA also cannot be generated through FoG, raising concerns about whether the performance of the proposed framework is dependent on METRA.
* Although the paper explains why [2] was not used as a baseline, the lack of comparison with other studies that utilize foundation models appears to be a weakness of the paper.

[1] Xie, Tianbao, et al. "Text2reward: Automated dense reward function generation for reinforcement learning." arXiv preprint arXiv:2309.11489 (2023).
[2] Rho, Seungeun, et al. "Language Guided Skill Discovery." arXiv preprint arXiv:2406.06615 (2024).

**Questions:**

* A comparison with other studies that utilize foundation models appears to be necessary.
* The paper mentioned in the Weakness section, [2], claims to be able to discover useful skills in environments utilizing robotic arms. In contrast, FoG appears to have failed to discover useful skills in the Franka Kitchen environment. I'm curious what differences between the two methods led to these contrasting results.

[2] Rho, Seungeun, et al. "Language Guided Skill Discovery." arXiv preprint arXiv:2406.06615 (2024).

---

> ### Author Response · Authors · 2024-11-22
> **Author Response (22/11/24)**
>
> Thank you for your review!
>
> > Studies such as [1] have already researched generating reward functions using foundation models. While there is some differentiation in applying a similar idea to skill discovery, the novelty appears limited compared to previous research that generated entire rewards through foundation models. The proposed idea only utilizes foundation models in the restricted aspect of 'applying weights to rewards proposed in other studies'.
>
> We agree that previous work already leverage foundation models to generate reward functions to train RL agents, such as [1, 3]. However, most of them only work for state-based input. For pixel-based tasks, previous work [4] shows that foundation models can only output very noisy rewards which can not be used to directly train RL agents. In our work, we actually show that such “noisy” signals can come into play in the skill discovery field.
>
> > It seems that skills that cannot be generated by METRA also cannot be generated through FoG, raising concerns about whether the performance of the proposed framework is dependent on METRA.
>
> Yes, we think this might be the case. However, for example, in Fig. 6, we did see that FoG learns to move different directions with METRA, suggesting that FoG might have potential to discover different skills with the underlying used skill discovery algorithms.
>
> > Although the paper explains why [2] was not used as a baseline, the lack of comparison with other studies that utilize foundation models appears to be a weakness of the paper.
>
> Thanks for pointing out. We now added two more baselines, METRA+ which uses human-defined score functions, and DoDont+ which uses foundation models annotated demonstrations for training DoDont. Please see updated Fig.4 and Fig.5.
>
> >The paper mentioned in the Weakness section, [2], claims to be able to discover useful skills in environments utilizing robotic arms. In contrast, FoG appears to have failed to discover useful skills in the Franka Kitchen environment. I'm curious what differences between the two methods led to these contrasting results.
>
> First, [2] is using state-based input, while FoG takes pixel-based input, which makes them not directly comparable. Meanwhile, FoG’s performance largely depends on the performance of the underlying used visual foundation models. For Franka Kitchen, it appears that CLIP cannot distinguish different visual states well, resulting in poor performance. With stronger foundation models that are more accurate, we believe the performance of FoG can be enhanced.
>
> ------------
> [1] Xie, Tianbao, et al. "Text2reward: Automated dense reward function generation for reinforcement learning." arXiv preprint arXiv:2309.11489 (2023).
>
> [2] Rho, Seungeun, et al. "Language Guided Skill Discovery." arXiv preprint arXiv:2406.06615 (2024).
>
> [3] Ma, Yecheng Jason, et al. "Eureka: Human-Level Reward Design via Coding Large Language Models." arXiv preprint arXiv: Arxiv-2310.12931
>
> [4] Adeniji, Ademi, et al. "Language reward modulation for pretraining reinforcement learning." arXiv preprint arXiv:2308.12270 (2023).

---

> > ### Comment · Area_Chair_W1v2 · 2024-11-24
> > **Please respond to rebuttal ASAP**
> >
> > Dear reviewer,
> > The process only works if we engage in discussion. Can you please respond to the rebuttal provided by the authors ASAP?

---

> > ### Comment · Reviewer_KNUe · 2024-11-25
> > **Official Comment by Reviewer KNUe**
> >
> > Thank you to the authors for their responses. Through additional comparisons with baselines utilizing foundation models, we were able to confirm the distinctiveness of the proposed framework. However, there are still some unclear points.
> >
> > - The authors mentioned, “such ‘noisy’ signals can come into play in the skill discovery field.” Does this imply that errors can occur in the process of generating overall rewards using the foundation model, and thus you applied a conservative approach by assigning weights to the skill discovery reward to minimize issues arising from the foundation model?
> >
> > - Additionally, you stated, “First, [2] is using state-based input, while FoG takes pixel-based input, which makes them not directly comparable.” As I understand it, FoG is not specifically designed for pixel-based input and is applicable to state-based input as well. Does this mean that the experiment conducted in Appendix A.6 used pixel-based input, making direct comparison with [2] difficult?

---

> > > ### Author Response · Authors · 2024-11-25
> > > **Author Response (25/11/24)**
> > >
> > > Thanks for your responses.
> > >
> > > > The authors mentioned, “such ‘noisy’ signals can come into play in the skill discovery field.” Does this imply that errors can occur in the process of generating overall rewards using the foundation model, and thus you applied a conservative approach by assigning weights to the skill discovery reward to minimize issues arising from the foundation model?
> > >
> > > Yes, we recognize that the foundation model we used (CLIP) performs better at answering simple "yes" or "no" questions rather than providing detailed reward outputs. Consequently, we applied it in a way that requires only binary responses. For example, it answers "yes" if the Cheetah does not flip and "no" if it does, and we then use these responses for weighting. This approach can indeed be considered a more "conservative" method.
> > >
> > > > Additionally, you stated, “First, [2] is using state-based input, while FoG takes pixel-based input, which makes them not directly comparable.” As I understand it, FoG is not specifically designed for pixel-based input and is applicable to state-based input as well. Does this mean that the experiment conducted in Appendix A.6 used pixel-based input, making direct comparison with [2] difficult?
> > >
> > > Yes, you are correct that FoG works with both state-based and pixel-based inputs. The experiments in Appendix A.6 on the Franka Kitchen environment indeed use pixel-based inputs, whereas [2] uses state-based inputs for Franka robot-arm object manipulation tasks. From the perspective of input types, directly comparing the two would be unfair. Additionally, the Franka robot-arm object manipulation task ([2]) is different from the Franka Kitchen task used by FoG.
> > >
> > >
> > > --------
> > >
> > > [2] Rho, Seungeun, et al. "Language Guided Skill Discovery." arXiv preprint arXiv:2406.06615 (2024).

---

### Official Review · Reviewer_RxEG · 2024-11-01

**Soundness:** 2
**Presentation:** 3
**Contribution:** 1
**Rating:** 3
**Confidence:** 5

**Summary:**

This paper presents a method called Foundation model Guided (FoG) skill discovery, which uses foundation models to incorporate human intentions into the process of skill discovery in reinforcement learning. FoG aims to align skills with specified human intentions, assigning higher rewards to desirable states and lower ones to undesirable states. It applies a score function generated by an LLM for state-based RL or implemented by CLIP for pixel-based RL to re-weight intrinsic rewards, guiding agents to avoid hazardous areas or eliminate undesirable behaviors. FoG is evaluated against baselines like METRA and DoDont, with experiments showing improved avoidance of undesirable behaviors and hazardous areas in specific environments.

**Strengths:**

The paper is generally well-organized, and the description of the FoG approach and its score function is detailed.

**Weaknesses:**

**The methodology is the same as DoDont.**

The core methodology of FoG closely resembles the DoDont approach, as both employ a score function derived from human-provided preferences, especially the form $\max\sum p(s,s') \|\phi(s') - \phi(s)\| z, \quad\text{s.t.} \|\phi(s')-\phi(s)\|\le1 $. Although FoG uses $f(s')$ in its objective, i.e., $\sum f(s') \|\phi(s') - \phi(s)\| z, \quad\text{s.t.} \|\phi(s')-\phi(s)\|\le1$, $f(s')$ is just a special case of $p(s,s')$. The similarity to DoDont weakens FoG's novelty. The main contribution of the paper is thus this special implementation of the score function proposed by DoDont. However, as stated below, even this special implementation is trivial.

**The score function is trivial, and using the vision language model as the score function to capture the human intention for MuJoCo robots has been proposed.**

For state-based RL, FoG utilizes ChatGPT to "write" a function to indicate if a certain dimension of the state is larger than a threshold; for instance,
```python
f(s) = lambda s: s[1] > threshold
```
This function is a trivial implementation of the score function, i.e., $p(s,s') = $ I(s'[1] > threshold), i.e., "s'[1] > threshold" is classified as "desired" and "s'[1] <= threshold" labeled as "undesired".

For pixel-based RL, FoG also utilizes the CLIP similarity score to implement a trivial score function $p(s,s') =$ CLIP(s', "desired") > CLIP(s', "undesired"). For instance, FoG utilizes CLIP(s', "ground is blue") > CLIP(s', "ground is orange") to encourage the halfcheetah to appear in the blue ground. However, one can easily obtain the visual state of the halfcheetah in the blue ground and the visual state of the halfcheetah in the orange ground and label the former as the "desired" state and the latter as the "undesired" state to train an image classifier for DoDont. The training process of the image classifier in DoDont can be regarded as a distillation from the pre-trained CLIP or human knowledge to a lightweight neural classifier. Similarly, the descriptions of "flip over" vs. "stand" and "twist" vs. "stretch" can easily be distilled from CLIP, other foundation models, or even human annotators. What's more, when there is no color in the ground or the color distribution in the left plane is the same as the right plane, the CLIP image encoder would fail since it does not capture any feature related to the motion.

Additionally, utilizing the vision language model to capture the human intention has been proposed by Juan et al.^1.

1. Juan Rocamonde, Victoriano Montesinos, Elvis Nava, Ethan Perez, David Lindner, Vision-Language Models are Zero-Shot Reward Models for Reinforcement Learning, ICLR 2024.


**Baseline comparison is unfair.**

The comparison to METRA is unconvincing because METRA does not incorporate any human intentions or extrinsic reward functions. For a fair evaluation, FoG should be compared with baselines that also incorporate human intentions, for instance, at least METRA+ implemented in DoDont, and might also consider other baselines utilized in DoDont, for instance,  DGPO and SMERL.

**The experiments lack thoroughness, especially regarding visualizing score functions.**

For Section 4.2, the paper could benefit from visual comparisons between the score functions utilized by FoG and DoDont in pixel-based Cheetah and pixel-based Quadruped. As stated in Line 332, in Cheetah, FoG uses only simple descriptions like "ground is blue" and "ground is orange" to signal whether the agent is on the left or right part. This is actually as simple as "x<0" and "x>0" as long as the image encoder, such as the pre-trained CLIP image encoder, can differentiate between blue and orange. Thus, the explanation for the "failure" of the DoDont method in Cheetah should at least include the comparison between the performance of the score function of the DoDont, which is an image classifier similar to the CLIP image encoder. In the implementation details mentioned in Appendix A.9.2, the authors only mention that the image classifier utilized in DoDont is trained by themselves. No further details are provided. The implementation details of the score function are critical since both DoDont and FoG are based on METRA.

For Section 4.3, I'm afraid I have to disagree with the authors that  "twisted" or "stretched" samples are hard to obtain. Since both "twist" and "stretch" are relatively stable gestures, one can easily obtain such gestures via the MuJoCo UI provided by the Deepmind Control Suite and take a screenshot. Even without UI and without human resources to label, one can instruct a VLM  to score each or every N-th visual state with the same intention, i.e., either the agent is stretched or the agent is twisted, and then use the labeled batch as the demonstration to train the score function.

**Questions:**

The description in Figure 5 is somewhat confusing. From the top subfigure, it is evident that MATRA, DoDont, and FoG all occupy both sides of the plane, with trajectories distributed on both the left and right. However, the authors assert that FoG avoids the right side, which is inconsistent with the experimental results. Please clarify this.

---

> ### Author Response · Authors · 2024-11-22
> **Author Response Part 1 of 2 (22/11/24)**
>
> Thank you for your review!
> > The methodology is the same as DoDont.
>
> FoG is not only closely related to DoDont, but also close to LSD, CSD, METRA. As can be seen in Table 1, all of these methods (including FoG) use the same objective (Distance-maximizing Skill Discovery objective proposed by CSD), but leverage different underlying distance metrics, which have different properties.
>
> > For state-based RL, FoG utilizes ChatGPT to "write" a function to indicate if a certain dimension of the state is larger than a threshold; for instance, ```f(s) = lambda s: s[1] > threshold```. This function is a trivial implementation of the score function, i.e.,  I(s'[1] > threshold), i.e., "s'[1] > threshold" is classified as "desired" and "s'[1] <= threshold" labeled as "undesired".
>
> The intention of FoG is to show that foundation models can be used to guide skill discovery in both state and pixel-based tasks, instead of showing how complicated score fn that foundation models can propose. Meanwhile, such a score function might seem too trivial to design for humans, it actually requires foundation models to understand 1) our intention; 2) identify the important state dimensions; 3) propose the logic of implementation, etc. which is not trivial for computers to do.
>
> > For pixel-based RL, FoG also utilizes the CLIP similarity score to implement a trivial score function  CLIP(s', "desired") > CLIP(s', "undesired"). For instance, FoG utilizes CLIP(s', "ground is blue") > CLIP(s', "ground is orange") to encourage the halfcheetah to appear in the blue ground. However, one can easily obtain the visual state of the halfcheetah in the blue ground and the visual state of the halfcheetah in the orange ground and label the former as the "desired" state and the latter as the "undesired" state to train an image classifier for DoDont. The training process of the image classifier in DoDont can be regarded as a distillation from the pre-trained CLIP or human knowledge to a lightweight neural classifier. Similarly, the descriptions of "flip over" vs. "stand" and "twist" vs. "stretch" can easily be distilled from CLIP, other foundation models, or even human annotators.
>
> Thanks for pointing out. We added the DoDont trained by CLIP annotated demonstrations as a baseline (DoDont+). Turns out DoDont+ does not really work very well. See updated Fig. 5, DoDont with expert demonstrations can learn to avoid bottom-left area in Quadruped. Then we annotate these demonstrations using CLIP (what FoG uses) and use the annotated demonstrations to train DoDont+. Since CLIP is not able to 100% annotate “dos” and “donts”, the demos end up with ~70% of accuracy. Then such inaccuracy is injected to the classifier of DoDont+, which consistently provides inaccurate signals, resulting in poor performance. Whereas FoG uses CLIP on the fly, it has ~70% accuracy, the overall trend is positive, leading to a self-correct system and ending up with better performance.
>
> > What's more, when there is no color in the ground or the color distribution in the left plane is the same as the right plane, the CLIP image encoder would fail since it does not capture any feature related to the motion.
>
> We totally agree with this point, and this is a problem of all these Distance-maximizing Skill Discovery methods [1]. In such a case, none of FoG, DoDont nor METRA would work using pixel-input.  We don't think it's suitable to use a vision-based foundation model when there are no salient features in the image to distinguish between states. CLIP is best used in settings with clear visual features, like Cheetah, Quadruped with colored planes. Note that we used these environments out of the box from prior work [1], and did not do any additional modifications to make CLIP better.
>
> > Additionally, utilizing the vision language model to capture the human intention has been proposed by Juan et al.^1.
>
> [2] indeed shows that vlm can be used to capture the human intention to provide step-wise reward signals to RL agents. But they are too noisy to directly train the agents (36% success rate). In order to make them work, authors need to 1) change the texture of the environment background to be more realistic; 2) change the camera view. In contrast, FoG provides an idea of how we can leverage vlm and avoid outputting such noisy step-wise signals to better guide skill discovery without any specific modifications of the original environments.
>
> ------------------
>
> [1] Park, Seohong, Oleh Rybkin, and Sergey Levine. "Metra: Scalable unsupervised rl with metric-aware abstraction." arXiv preprint arXiv:2310.08887 (2023).
>
> [2] Rocamonde, Juan, et al. "Vision-language models are zero-shot reward models for reinforcement learning." arXiv preprint arXiv:2310.12921 (2023).

---

> > ### Author Response · Authors · 2024-11-22
> > **Author Response Part 2 of 2 (22/11/24)**
> >
> > > The comparison to METRA is unconvincing because METRA does not incorporate any human intentions or extrinsic reward functions. For a fair evaluation, FoG should be compared with baselines that also incorporate human intentions, for instance, at least METRA+ implemented in DoDont, and might also consider other baselines utilized in DoDont, for instance, DGPO and SMERL.
> >
> > Thanks for the suggestion. We now added two more baseline (METRA+ and DoDont+) to the results. The results suggest that if “perfect” reward functions or demonstrations are available, then METRA+ or DoDont can outperform. The advantage of FoG will start shining when these ingredients are difficult to obtain. For other baselines used in DoDont like DGPO and SMERL, we saw they perform poorly so we decided to not include them.
> >
> > > For Section 4.2, the paper could benefit from visual comparisons between the score functions utilized by FoG and DoDont in pixel-based Cheetah and pixel-based Quadruped. As stated in Line 332, in Cheetah, FoG uses only simple descriptions like "ground is blue" and "ground is orange" to signal whether the agent is on the left or right part. This is actually as simple as "x<0" and "x>0" as long as the image encoder, such as the pre-trained CLIP image encoder, can differentiate between blue and orange. Thus, the explanation for the "failure" of the DoDont method in Cheetah should at least include the comparison between the performance of the score function of the DoDont, which is an image classifier similar to the CLIP image encoder.
> >
> > We train the classifier in DoDont to have more than 97% accuracy on the demonstrations. Then we test the classifier on the videos that are not used for training the classifier. See the visualization / videos of output scores on our project website (https://sites.google.com/view/iclr-fog). In Cheetah flip experiments, the classifier seems to exploit the color of the ground (as also explained in the DoDont paper), outputting high scores for the flip posture not in the training demos. In Cheetah avoid experiments, the classifier outputs higher scores when the agent is going left, and outputs lower scores when the agent is going right, which is aligned with the training objective. But weirdly, It does not learn to avoid the hazardous areas (right areas), and we are still investigating.
> >
> > > In the implementation details mentioned in Appendix A.9.2, the authors only mention that the image classifier utilized in DoDont is trained by themselves. No further details are provided. The implementation details of the score function are critical since both DoDont and FoG are based on METRA.
> >
> > Thanks for pointing it out. We now updated more details about DoDont and include the demonstrations we used for training DoDont on the project website (https://sites.google.com/view/iclr-fog). We train classifiers in DoDont to always have more than 95% of accuracy and then stop.
> >
> > > For Section 4.3, I'm afraid I have to disagree with the authors that "twisted" or "stretched" samples are hard to obtain. Since both "twist" and "stretch" are relatively stable gestures, one can easily obtain such gestures via the MuJoCo UI provided by the Deepmind Control Suite and take a screenshot. Even without UI and without human resources to label, one can instruct a VLM to score each or every N-th visual state with the same intention, i.e., either the agent is stretched or the agent is twisted, and then use the labeled batch as the demonstration to train the score function.
> >
> > We consider the process that needs human intervention as “hard” to obtain, because it is not scalable. While obtaining demonstrations for a single task, like "twist" or "stretch," might be manageable, scaling this process to 1000 tasks becomes challenging either manually or with the help of vlms. Meanwhile, using an imperfect vlm (CLIP in our case, it labels ~30% dos as donts in our DoDont+ experiments) to create demonstrations to train a classifier will end up with an inaccurate classifier, which is problematic during the training. See the DoDont+ results discussed above.
> >
> > > The description in Figure 5 is somewhat confusing. From the top subfigure, it is evident that MATRA, DoDont, and FoG all occupy both sides of the plane, with trajectories distributed on both the left and right. However, the authors assert that FoG avoids the right side, which is inconsistent with the experimental results. Please clarify this.
> >
> > Thanks for pointing it out. We updated the caption of Fig.5, and changed it to “FoG has a clear preference to go left”.

---

> > > ### Comment · Area_Chair_W1v2 · 2024-11-24
> > > **Please respond to rebuttal ASAP**
> > >
> > > Dear reviewer,
> > > The process only works if we engage in discussion. Can you please respond to the rebuttal provided by the authors ASAP?

---

> > > ### Comment · Reviewer_RxEG · 2024-11-25
> > > **Reply to Author Response Part 2 of 2**
> > >
> > > Thank you to the authors for their response. However, their reply does not fully address my concerns.
> > >
> > > 5. As noted by the authors, METRA+ and DoDont+ outperform FoG when safety-related rewards or demonstrations are available. Although the authors assert that FoG will excel when these elements are challenging to acquire, I believe that these elements are readily accessible in the current reinforcement learning tasks utilized in this paper.
> > >
> > > 6. The authors reported that the classifier can accurately determine whether the agent is moving left or right; therefore, DoDont should perform as expected. This indicates potential bugs in the implementations of DoDont and DoDont+. I recommend that the authors conduct a more thorough investigation into this unusual phenomenon.
> > >
> > > 7. Scaling this process to 1,000 tasks presents challenges for both the baseline methods and FoG. Additionally, FoG has not provided explicit techniques or justifications to address the imperfectness of VLM; instead, similar to [2], FoG is a direct application of VLM to integrate human intention.
> > >
> > > Considering these weaknesses, I will maintain my score.

---

> > > > ### Author Response · Authors · 2024-11-27
> > > > **Author Response Part 2 of 2 (27/11/24)**
> > > >
> > > > > 5. As noted by the authors, METRA+ and DoDont+ outperform FoG when safety-related rewards or demonstrations are available. Although the authors assert that FoG will excel when these elements are challenging to acquire, I believe that these elements are readily accessible in the current reinforcement learning tasks utilized in this paper.
> > > >
> > > > The reviewer states that “these elements are readily accessible in the current RL tasks utilized in this paper”, which we agree, but we think investigating alternative methods that work with less assumptions or less human efforts should also be important.
> > > >
> > > > > 6. The authors reported that the classifier can accurately determine whether the agent is moving left or right; therefore, DoDont should perform as expected. This indicates potential bugs in the implementations of DoDont and DoDont+. I recommend that the authors conduct a more thorough investigation into this unusual phenomenon.
> > > >
> > > > As can be seen in the bottom row of Fig. 5,  DoDont and DoDont+ do work in Quadruped. In Cheetah, we indeed used “frame_stack=3”, we apologize for that. Now we changed it to “frame_stack=1” and reran DoDont and DoDont+. It turns out they still do not work well. We add more visualizations on the website, and you can see the classifiers of DoDont are unsure about states at the beginning, which might harm the exploration, resulting in smaller state coverage.
> > > >
> > > > > 7. Scaling this process to 1,000 tasks presents challenges for both the baseline methods and FoG. Additionally, FoG has not provided explicit techniques or justifications to address the imperfectness of VLM; instead, similar to [2], FoG is a direct application of VLM to integrate human intention.
> > > >
> > > > As we mentioned in the previous responses, our core focus is not on skill discovery in isolation or addressing the imperfectness of vlm, rather we are interested in investigating **how foundation models can improve skill discovery**. We think the reviewer might misunderstand the focus of FoG.

---

> ### Comment · Reviewer_RxEG · 2024-11-25
> **Reply to Author Response Part 1 of 2**
>
> Thanks to the authors for their clarifications. However, these clarifications cannot fully address my concerns.
>
> 1. **Methodology and Novelty**:   Previous methods, such as LSD, CSD, METRA, and DoDont, introduced novel components like Lipschitz constraints, controllability-aware distances, metric-aware abstraction, and intrinsic reward weighting, respectively. In contrast, FoG only substitutes score function implementation in DoDont with pre-trained foundation models. Moreover, the implementation of the score function has been demonstrated by [2].
>
> 2. **Use of Foundation Models**:  Foundation models, such as ChatGPT and Claude, are proficient at generating complex code based on human instructions. Using them to develop a predefined score function, as in FoG, is unsurprising and arguably represents a straightforward application. Although the authors' response emphasizes the complexity of understanding intentions, identifying critical state dimensions, and implementing logic, these tasks fall well within the capabilities of these models. This diminishes the originality claimed in this aspect.
>
> 3. **Comparison to DoDont+**:  Although the authors added DoDont+ as a baseline, its implementation may only partially match that of FoG. Specifically, if demonstrations are "automatically" annotated by CLIP, then DoDont+ should similarly leverage CLIP to annotate all past training samples during training dynamically. An on-the-fly version of DoDont+ would align better with FoG's setup and provide a more equitable comparison. The current implementation may not adequately reflect the potential of DoDont+ with an automatic annotation model like CLIP, which undermines the validity of the performance comparison.
>
> 4. **Overlap with Related Work**:  The difference between FoG and prior work such as [2] remains unclear. Both approaches utilize Vision-Language Models (VLMs) to capture human intentions utilizing cosine similarity between visual and textual embeddings. I disagree with the authors that FoG does not modify the "original" MuJoCo environment since the environment used in FoG, followed by previous work like METRA, has already been modified. For instance, the floor color has been changed. Moreover, [2] utilizes the OpenAI MuJoCo environment while FoG adopts the DeepMind Control Suites. The noisy level of the gymnasium MuJoCo environment and DeepMind Control Suites is certainly different. It is well-known in the RL community that gymnasium MuJoCo and DeepMind Contro Suites have different dynamics. If the author argues that FoG is more stable than [2], they at least need to compare the reward function of FoG and [2] under the same simulation environments.

---

> ### Author Response · Authors · 2024-11-27
> **Author Response Part 1 of 2 (27/11/24)**
>
> Thanks for your response.
>
> > 1. Methodology and Novelty: Previous methods, such as LSD, CSD, METRA, and DoDont, introduced novel components like Lipschitz constraints, controllability-aware distances, metric-aware abstraction, and intrinsic reward weighting, respectively. In contrast, FoG only substitutes score function implementation in DoDont with pre-trained foundation models. Moreover, the implementation of the score function has been demonstrated by [2].
>
> The reviewer seems concerned about novelty, i.e. we are not contributing a novel component to skill discovery, or coming up with  a new score function for VLMs. However, our core focus is not on skill discovery in isolation, rather  we are interested in investigating **how foundation models can improve skill discovery**. Our solution is to weight the skills using a distance metric defined by the foundation model, and we believe this is a simple and effective way to incorporate foundation models into skill discovery, as shown by our results.
>
> > 2. Use of Foundation Models: Foundation models, such as ChatGPT and Claude, are proficient at generating complex code based on human instructions. Using them to develop a predefined score function, as in FoG, is unsurprising and arguably represents a straightforward application. Although the authors' response emphasizes the complexity of understanding intentions, identifying critical state dimensions, and implementing logic, these tasks fall well within the capabilities of these models. This diminishes the originality claimed in this aspect.
>
> As mentioned above, while it is possible for foundation models to solve tasks, they come with their own drawbacks, like prompting, resource efficiency at inference time, etc. We don’t think it’s a fair argument to claim that because foundation models could directly solve tasks, this diminishes the potential of alternative approaches.
>
> > 3. Comparison to DoDont+: Although the authors added DoDont+ as a baseline, its implementation may only partially match that of FoG. Specifically, if demonstrations are "automatically" annotated by CLIP, then DoDont+ should similarly leverage CLIP to annotate all past training samples during training dynamically. An on-the-fly version of DoDont+ would align better with FoG's setup and provide a more equitable comparison. The current implementation may not adequately reflect the potential of DoDont+ with an automatic annotation model like CLIP, which undermines the validity of the performance comparison.
>
> If we do what the reviewer described, we would think it is an entirely different algorithm, instead of a variant of DoDont. We think the core idea of DoDont is to use a pre-trained classifier that is trained on pre-collected demonstrations, instead of 1) update the classifier on-the-fly; 2) annotate the demonstrations using foundation models on-the-fly.
>
> > 4. Overlap with Related Work: The difference between FoG and prior work such as [2] remains unclear. Both approaches utilize Vision-Language Models (VLMs) to capture human intentions utilizing cosine similarity between visual and textual embeddings. I disagree with the authors that FoG does not modify the "original" MuJoCo environment since the environment used in FoG, followed by previous work like METRA, has already been modified. For instance, the floor color has been changed. Moreover, [2] utilizes the OpenAI MuJoCo environment while FoG adopts the DeepMind Control Suites. The noisy level of the gymnasium MuJoCo environment and DeepMind Control Suites is certainly different. It is well-known in the RL community that gymnasium MuJoCo and DeepMind Contro Suites have different dynamics. If the author argues that FoG is more stable than [2], they at least need to compare the reward function of FoG and [2] under the same simulation environments.
>
> In our previous response, we mentioned that “we used these environments out of the box from prior work [1], and did not do any additional modifications to make CLIP better.” We never claimed that we did not modify the “original” MuJoCo environment, what we meant was we did not modify environments we took from [1], which all other baselines are used, including DoDont (part of), METRA.
>
> —-------
>
> [1] Park, Seohong, Oleh Rybkin, and Sergey Levine. "Metra: Scalable unsupervised rl with metric-aware abstraction." arXiv preprint arXiv:2310.08887 (2023)

---

### Official Review · Reviewer_sdar · 2024-11-03

**Soundness:** 2
**Presentation:** 3
**Contribution:** 1
**Rating:** 5
**Confidence:** 3

**Summary:**

Existing skill discovery methods typically focus on learning to reach diverse states. However, they may learn undesirable behaviors and possibly dangerous skills. To this end, the author proposes to use LLM and CLIP to generate a score function that judge whether the current state is undesirable (according to given text description of desirable and undesirable behaviors).

**Strengths:**

The paper is well written and easy to follow. The experiments show that the proposed method successfully achieve the proposed goal -- learning diverse skills while avoiding undesirable behavior, on both object state and image space.

**Weaknesses:**

Though learning diverse skills while avoiding undesirable behaviors is a valid motivation, using foundation models simply as an undesirable behavior recoginizer seems to be a weird use case. In the case that we already use foundation models, it seem to me a more natural way is to let foundation models propose some tasks and safety reward (if applicable), examples include but are not limited to [1] and [2]. This allows skill discovery to directly target at learning more semantically-meaningful skills, compared to target at visiting diverse states.

Could authors providing the reasoning for this design choice? (though a bit off-topic, though the authors mentions that Eureka requires multiple iterations of reward designs, I am not fully conviced. Learning a skill for each reward during the iterations could also lead to diverse skills).

[1] Ma, Yecheng Jason, et al. "DrEureka: Language Model Guided Sim-To-Real Transfer." arXiv preprint arXiv:2406.01967 (2024).
[2] Zhao, Xufeng, Cornelius Weber, and Stefan Wermter. "Agentic Skill Discovery." arXiv preprint arXiv:2405.15019 (2024).

**Questions:**

none

---

> ### Author Response · Authors · 2024-11-22
> **Author Response (22/11/24)**
>
> Thank you for your review!
>
> > Though learning diverse skills while avoiding undesirable behaviors is a valid motivation, using foundation models simply as an undesirable behavior recoginizer seems to be a weird use case. In the case that we already use foundation models, it seem to me a more natural way is to let foundation models propose some tasks and safety reward (if applicable), examples include but are not limited to [1] and [2]. This allows skill discovery to directly target at learning more semantically-meaningful skills, compared to target at visiting diverse states.
>
> FoG works with both state-based and pixel-based input. In state-based tasks, we let foundation models output a score function that is similar to the “safety reward” that you mentioned. Similar to works you mentioned [1] and [2] both only work on state-based tasks.
>
> However, using foundation models output the reward for pixel-based tasks is not trivial, for example, [3,4] show that step-wise reward output by the foundation models are too noisy to directly train a RL agent. Thus, FoG leverages foundation models as “behavior recognizers” in unsupervised skill discovery instead of using them for providing step-wise reward signals directly.
>
> > Could authors providing the reasoning for this design choice? (though a bit off-topic, though the authors mentions that Eureka requires multiple iterations of reward designs, I am not fully conviced. Learning a skill for each reward during the iterations could also lead to diverse skills).
>
> Eureka [5] is set up under the conventional RL setting, where there is a reward fn and the agent optimizes the reward fn. At the end of the day, the agent learns to achieve the task that the reward fn is defined. In order to learn diverse skills by optimizing a predefined reward fn, it requires 1) knowing all skills in advance; 2) defining a reward function for each skill. FoG is under an unsupervised RL (URL) setting, where there’s no task-specific predefined reward function. In URL, 1)we don’t assume having access to the downstream tasks, all we want is to learn diverse skills, which could possibly include some downstream tasks we are interested in; 2) all skills are learned by optimizing a single unsupervised objective, which defines how diverse / distinguishable the skills are.
>
> —---------------
>
> [1] Ma, Y. J., et al. "DrEureka: Language model guided sim-to-real transfer. arXiv 2024." arXiv preprint arXiv:2406.01967.
>
> [2] Zhao, Xufeng, Cornelius Weber, and Stefan Wermter. "Agentic Skill Discovery." arXiv preprint arXiv:2405.15019 (2024).
>
> [3] Adeniji, Ademi, et al. "Language reward modulation for pretraining reinforcement learning." arXiv preprint arXiv:2308.12270 (2023).
>
> [4] Rocamonde, Juan, et al. "Vision-language models are zero-shot reward models for reinforcement learning." arXiv preprint arXiv:2310.12921 (2023).
>
> [5] Ma, Yecheng Jason, et al. "Eureka: Human-Level Reward Design via Coding Large Language Models." arXiv preprint arXiv: Arxiv-2310.12931

---

> > ### Comment · Area_Chair_W1v2 · 2024-11-24
> > **Please respond to rebuttal ASAP**
> >
> > Dear reviewer,
> > The process only works if we engage in discussion. Can you please respond to the rebuttal provided by the authors ASAP?

---

> > ### Comment · Reviewer_sdar · 2024-11-24
> >
> > Thanks for your detailed response! I have raised my score to 5. The reason that I didn't raise to 6 is that I feel the motivation (and thus contribution) of the paper still can be improved.
> > Personally, I think the major issue of unsupervised skill discovery is that most of the skills are useless (of course, some of them are not safe). So I feel more valuable directions of incorporating foundation models into skill discovery is guiding skill learning to better align with potential tasks, in contrast to simply avoiding region A or skill B (as there could be too many things to avoid).
> > Even though the current LLM/VLM-based method can't be directly applied to image-based environments to provide reward for each step, one potential workaround is to let them generate the prompt for CLIP and then let CLIP provide step-wise reward.

---

> > > ### Author Response · Authors · 2024-11-25
> > > **Author Response (25/11/24)**
> > >
> > > Thanks for your responses.
> > >
> > > > Personally, I think the major issue of unsupervised skill discovery is that most of the skills are useless (of course, some of them are not safe). So I feel more valuable directions of incorporating foundation models into skill discovery is guiding skill learning to better align with potential tasks, in contrast to simply avoiding region A or skill B (as there could be too many things to avoid). Even though the current LLM/VLM-based method can't be directly applied to image-based environments to provide reward for each step, one potential workaround is to let them generate the prompt for CLIP and then let CLIP provide step-wise reward.
> > >
> > > Although FoG does not explicitly try to align skill discovery with the downstream tasks, we show that by learning desirable skills, FoG achieves better downstream task performance than other baselines. See appendix A.3, the downstream task performance on Cheetah. FoG eliminates flipping and results in better running performance.
> > >
> > > We agree that aligning skill discovery with potential downstream tasks would be an interesting and important direction to chase, however, it is out of the scope of this work.

---

### Official Review · Reviewer_pybd · 2024-11-03

**Soundness:** 3
**Presentation:** 3
**Contribution:** 2
**Rating:** 5
**Confidence:** 4

**Summary:**

The paper presents a foundation model-based approach for guiding unsupervised skill discovery to discover desirable skills and avoid undesirable skills. The key idea of the method is to encode human intentions into skill discovery using foundation models, by extracting a representative score function from them. By using these scores to re-weight the rewards of skill discovery methods, the approach achieves skills aligned with human intentions, while avoiding those that are misaligned.

**Strengths:**

Generally, the paper is well-written and the underlying idea is fairly straightforward. The authors have adopted the METRA framework, and can thus handle image inputs

**Weaknesses:**

In terms of the claimed novelties, one of them is that FoG can learn behaviors that are challenging to define. I do not believe this is a new phenomenon – it is fairly well documented in preference based RL works. In the same light, I believe the work has overlooked some very relevant works on safe/guided skill discovery which I believe could be used as baselines or at least discussed in detail.

**Questions:**

Generally, the paper is well-written and the underlying idea is fairly straightforward. In terms of the claimed novelties, one of them is that FoG can learn behaviors that are challenging to define. I do not believe this is a new phenomenon – it is fairly well documented in preference based RL works [1]. In the same light, I believe the work have overlooked some very relevant works [2,3] which I believe could be used as baselines or at least discussed in detail. My detailed comments are as follows:

1.	How does the method related to papers [2] and [3]? Does FoG provide any specific advantages over these methods? If so, would it be fair to consider these are relevant baselines for comparison?
2.	In lines 251-252, how did the foundation models develop this insight about the angle of the cheetah’s front tip? On a related note, can we guarantee the score functions are always appropriate?
3.	Is it assumed that the score can be captured purely by virtue of observing s’ (Eq 6)? Do these scores not consider the manner in which a task is achieved? For example, if the task is to cut a fruit, this could be done with a kitchen knife or a hacksaw. Although the final outcome (the fruit cut into pieces) may be identical, the way it is done also involves some alignment signals – does the proposed score function capture such process-based alignment (which would be based on state trajectories, and not just a single state) as opposed to state/outcome based alignment?
4.	Regarding lines 160-161, is the score function always binary in nature? If so, I wonder whether setting $\alpha>0$ even matters as long as $\alpha<1$.
5.	The experiments are run over just 3 seeds, which is quite low. Is there any specific reason for this?
6.	In the experiment relating to Fig 6, is the task to move while being stretched/twisted? If it is just to be in stretched/twisted positions, why do videos need to be shown?
7.	It is possible that in the video, the agent is initially stretched and then towards the end, gets twisted. How would this be handled?
8.	It would be better to include the full context/descriptions used to generate the score function.
9.	Typos- line 244, 257-258, 395



[1] Christiano, Paul F., et al. "Deep reinforcement learning from human preferences." Advances in neural information processing systems 30 (2017).

[2] Kim, Sunin, et al. "Safety-aware unsupervised skill discovery." 2023 IEEE International Conference on Robotics and Automation (ICRA). IEEE, 2023.

[3] Hussonnois, Maxence, , et al. "Controlled Diversity with Preference: Towards Learning a Diverse Set of Desired Skills." Proceedings of the 2023 International Conference on Autonomous Agents and Multiagent Systems. 2023.

---

> ### Author Response · Authors · 2024-11-22
> **Author Response Part 1 of 2 (22/11/24)**
>
> Thank you for your review!
> >1. How does the method related to papers [2] and [3]? Does FoG provide any specific advantages over these methods? If so, would it be fair to consider these are relevant baselines for comparison?
>
> Thank you for pointing out [2] and [3]. They are indeed quite related, as they both integrate human intentions into skill discovery. However, both of them only work in state-based tasks, while FoG works in both state-based and pixel-based tasks. Both [2] and [3] require humans to design either safety-rule or provide preferences, while FoG leverages foundation models to do so, which is scalable.
>
> [2] integrates a safety-indicator function into the Critic learning of the unsupervised skill discovery method to achieve learning safe behaviors. However, the safety-indicator function needs to be pre-defined. It is similar to our newly added baseline METRA+, which uses a predefined score function. The problem with these kinds of pre-defined score functions / reward functions is that: if it is perfectly defined, the agent will learn well (see updated Fig.6, METRA+ can learn to perfectly avoid hazardous areas). If not, which is generally the case, (see updated Fig.5, METRA+ cannot learn to avoid flipping using a simple predefined score function), the agent might struggle.
>
> [3] leverages preference-based RL to integrate human’s preference, it requires human-in-the-loop to continuously provide preference on trajectory segments. In more complex tasks where the training time is long, this might be infeasible.
>
> >2.In lines 251-252, how did the foundation models develop this insight about the angle of the cheetah’s front tip? On a related note, can we guarantee the score functions are always appropriate?
>
> For state-based tasks, we input the task description and state space (basically most information you can see on the Gymnasium webpage, for example: https://gymnasium.farama.org/environments/mujoco/ant/) to the foundation model as the context. But we do NOT tell foundation models which dimension they should focus on or look at specifically.
>
> In state-based tasks, we cannot 100% guarantee that the score functions are always appropriate. However, during the experiments, we tried different foundation models (ChatGPT, Claude) on different state-based tasks (ant and half-cheetah), and it seems both of them can give us reasonable score functions on both these two tasks. Our intuition is that since all the context is quite clear and presented, and the task is relatively easy, the state-of-the-art foundation models can easily recognise the needed information. If the task gets more complicated, we might need to take an iterative way to gradually guide function models to output the appropriate score function like what Eureka [1] does.
>
> >3. Is it assumed that the score can be captured purely by virtue of observing s’ (Eq 6)? Do these scores not consider the manner in which a task is achieved? For example, if the task is to cut a fruit, this could be done with a kitchen knife or a hacksaw. Although the final outcome (the fruit cut into pieces) may be identical, the way it is done also involves some alignment signals – does the proposed score function capture such process-based alignment (which would be based on state trajectories, and not just a single state) as opposed to state/outcome based alignment?
>
> Indeed, FoG aims to provide step-wise reward signals, thus, the score fn is based on states. Current FoG does not capture the process-based alignment, but in future work, we mentioned FoG has the potential to integrate multiple intentions simultaneously, in your case, such process-based alignment could be potentially achieved by using two intentions, for example, “the agent is using the knife” and “the fruit is cut into pieces”.
>
> —---------------
>
> [1] Ma, Yecheng Jason, et al. "Eureka: Human-Level Reward Design via Coding Large Language Models." arXiv preprint arXiv: Arxiv-2310.12931
>
> [2] Kim, Sunin, et al. "Safety-aware unsupervised skill discovery." 2023 IEEE International Conference on Robotics and Automation (ICRA). IEEE, 2023.
>
> [3] Hussonnois, Maxence, , et al. "Controlled Diversity with Preference: Towards Learning a Diverse Set of Desired Skills." Proceedings of the 2023 International Conference on Autonomous Agents and Multiagent Systems. 2023.

---

> > ### Author Response · Authors · 2024-11-22
> > **Author Response Part 2 of 2 (22/11/24)**
> >
> > >4. Regarding lines 160-161, is the score function always binary in nature? If so, I wonder whether setting $\alpha>0$ even matters as long as $\alpha<1$.
> >
> > The idea of the score function is to output higher scores if the state is desirable and lower scores if the state is undesirable, so it does not necessarily always have to be binary (0 or 1). Setting $\alpha$ to different values ($0< \alpha <1$) affects the performance differently, please see Figure 7.
> >
> > >5. The experiments are run over just 3 seeds, which is quite low. Is there any specific reason for this?
> >
> > We found FoG is pretty robust across different seeds, as can be seen in Figure 4, 5, 6. So out of computation consideration, we decided to run over 3 seeds.
> >
> > >6. In the experiment relating to Fig 6, is the task to move while being stretched/twisted? If it is just to be in stretched/twisted positions, why do videos need to be shown?
> >
> > Yes, the humanoid robot needs to learn to move and be twisted.
> >
> > >7. It is possible that in the video, the agent is initially stretched and then towards the end, gets twisted. How would this be handled?
> >
> > The snapshots in Fig.6 are randomly taken, not the one at the end time step. So they should be able to provide an overview of humanoids’ postures across the whole process.
> >
> > >8. It would be better to include the full context/descriptions used to generate the score function.
> >
> > Yes, we include all of them in the appendix. Please see A.8 and A.12.
> >
> > >9. Typos- line 244, 257-258, 395
> >
> > Thanks for pointing this out. We have fixed these typos.

---

> > > ### Comment · Reviewer_pybd · 2024-11-24
> > >
> > > Thanks for these responses. In my opinion, 3 seeds is too few especially if computational costs are not a major concern. If FoG is indeed robust, what is the reason for this? It would be good to further investigate this aspect, since it could be of interest. I would recommend using more seeds anyway.
> > >
> > > The authors mention the humanoid needs to learn to move and be twisted - I want to clarify whether you mean it needs to learn to move while being in a twisted position?
> > >
> > > Thanks for the other responses.

---

> > > > ### Author Response · Authors · 2024-11-25
> > > > **Author Response Part 2 of 2 (25/11/24)**
> > > >
> > > > Thanks for your responses.
> > > >
> > > > > In my opinion, 3 seeds is too few especially if computational costs are not a major concern. If FoG is indeed robust, what is the reason for this? It would be good to further investigate this aspect, since it could be of interest. I would recommend using more seeds anyway.
> > > >
> > > > The robustness of FoG can be attributed to two key factors: 1) the underlying skill discovery method, METRA, on which FoG is built, is robust, i.e. it can steadily discover diverse and distinguishable skills; 2) foundation models used by FoG are frozen, allowing them to output relatively consistent results.  However, we agree with the reviewer that 3 seeds is a bit too few. We are currently running 2 additional seeds, bringing the total to five seeds upon completion.
> > > >
> > > > > The authors mention the humanoid needs to learn to move and be twisted - I want to clarify whether you mean it needs to learn to move while being in a twisted position?
> > > >
> > > > Yes, the agent needs to learn to move while being in a twisted position.

---

> > > > > ### Comment · Reviewer_pybd · 2024-11-27
> > > > >
> > > > > Thanks for your responses. Just to follow up - is it possible/straightforward to integrate FoG with non-METRA based methods for state-based tasks? If so, I wonder how this would perform relative to the discussed baselines. Also, it may serve as an ablation to show that the robustness indeed comes from METRA.

---

> > ### Comment · Reviewer_pybd · 2024-11-24
> >
> > Thanks for your responses. I understand the method's distinction from [2] and [3], but to me, they still seem very related. [3] does require a human in the loop, but I guess the method could easily apply with an LLM in the loop as well. At the very least, I believe these works should be discussed in detail, clearly describing the differences of these methods with the proposed one.  I would also include direct comparisons for state-based environments, but I am aware that may not be possible at this stage.
> >
> > It is important to clearly mention your intuitions and assumptions, such as the one in your response to 2. Process based alignment (mentioned in your response to 3.) could also be a current limitation. I would encourage the authors to list out all such assumptions and limitations (as well as potential solutions as suggested) in a separate "Limitations" section in the interest of providing readers with a complete picture of the contributions.

---

> > > ### Author Response · Authors · 2024-11-25
> > > **Author Response Part 1 of 2 (25/11/24)**
> > >
> > > Thanks for your responses.
> > > > I understand the method's distinction from [2] and [3], but to me, they still seem very related. [3] does require a human in the loop, but I guess the method could easily apply with an LLM in the loop as well. At the very least, I believe these works should be discussed in detail, clearly describing the differences of these methods with the proposed one.
> > >
> > > Another difference of [2,3] is that they are based on skill discovery methods that maximize the mutual information, which is shown to not always encourage the agent to discover distant states, as the mutual information objective can be satisfied by learning simple and static skills [1,4]. Ones we mainly discussed in the previous related work,  i.e. works that maximize Distance-maximizing Skill Discovery objective, that learns diverse skills while maximizing the traveled distance under the given distance metric. But we agree with you that these two are indeed very related and now discussed [2,3] in detail in the updated related work. Please see the updated manuscript.
> > >
> > > > It is important to clearly mention your intuitions and assumptions, such as the one in your response to 2. Process based alignment (mentioned in your response to 3.) could also be a current limitation. I would encourage the authors to list out all such assumptions and limitations (as well as potential solutions as suggested) in a separate "Limitations" section in the interest of providing readers with a complete picture of the contributions.
> > >
> > > Thanks for pointing it out. We now included a separate “Limitations” section in the updated manuscript, including all limitations we discussed here and ones in the previous future work.
> > >
> > > --------
> > > [1] Park, Seohong, Oleh Rybkin, and Sergey Levine. "Metra: Scalable unsupervised rl with metric-aware abstraction." arXiv preprint arXiv:2310.08887 (2023).
> > >
> > > [2] Kim, Sunin, et al. "Safety-aware unsupervised skill discovery." 2023 IEEE International Conference on Robotics and Automation (ICRA). IEEE, 2023.
> > >
> > > [3] Hussonnois, Maxence, et al. "Controlled Diversity with Preference: Towards Learning a Diverse Set of Desired Skills." Proceedings of the 2023 International Conference on Autonomous Agents and Multiagent Systems. 2023.
> > >
> > > [4] Park, Seohong, et al. "Lipschitz-constrained unsupervised skill discovery." International Conference on Learning Representations. 2022.

---

### Author Response · Authors · 2024-11-22
**Global Response (tl;dr of some common items & experiment update) (22/11/24)**

We would like to thank all the reviewers for their comments and efforts towards evaluating our paper.

In this post, we would like to provide a tl;dr summary for some key items of interest to multiple reviewers.

- **Lack of baselines (Reviewer pybd, RxEG, KNUe)**: We added two more baselines that incorporate human intentions by 1) using hand-crafted score functions (METRA+) and 2) leveraging foundation model labeled demonstrations (DoDont+). METRA+ (suggested by RxEG) uses human defined score functions, and DoDont+ (mentioned by RxEG) instead of using expert-level demonstrations, uses demonstrations that are labeled by foundation models (not 100% accurate). Results suggest that if a perfect score function can be designed or expert-level demonstrations can be obtained, METRA+ or DoDont can potentially outperform. However, FoG shines in scenarios where these ingredients are challenging to have, which is generally the case.

- **Motivation of why do not use foundation models directly for reward generation (Reviewer sdar, KNUe)**: Most previous methods that use foundation models to directly generate task reward functions only work for state-based tasks [1]. Using foundation models to provide step-wise reward signals in pixel-based tasks is too noisy to directly train RL agents [2]. FoG demonstrates that such noisy signals can come into play to guide skill discovery.

We updated the manuscript and highlighted the modified part with red colour. More details on each of these points are in the responses to individual reviewers.

-------------
[1] Ma, Yecheng Jason, et al. "Eureka: Human-Level Reward Design via Coding Large Language Models." arXiv preprint arXiv: Arxiv-2310.12931

[2] Rocamonde, Juan, et al. "Vision-language models are zero-shot reward models for reinforcement learning." arXiv preprint arXiv:2310.12921 (2023).

---

### Meta-Review · Area_Chair_W1v2 · 2024-12-20

**Metareview:**

The methodology is trying to incorporate foundation model guidance into guiding unsupervised skill discovery methods. The key idea is to have a foundation model write a score that is used to multiply with skill discovery rewards. This directs exploration away from "undesired" behavior and learns more interesting, hard to define skills. This work builds directly on top of METRA.

Strengths:
Results seem compelling in generating safer and interesting skills.
Methodology seems simple and easy to implement, works from pixels

Weaknesses:
The work is not very novel over baselines - eg DoDont, and can even be instantiated as a relatively simple version of the DoDont framework.
The work is not quite "unsupervised" anymore, and straddles the line between using foundation models for reward generation and unsupervised skill learning. It's not super obvious why we wouldn't just a foundation model to generate a bunch of different behaviors and corresponding rewards. This seems like a pretty strong baseline to compare with.

Overall, the last mentioned weakness above is the kicker. If the authors claim that unsupervised skill discovery is still necessary over just task generation + reward generation directly from a foundation model, this needs to be justified more clearly theoretically or empirically. Otherwise this baseline needs to be compared against. The argument about pixels vs state I think is tricky because it's not really that unsupervised RL methods are used for real-world robots either and are only used in simulation.

**Additional Comments On Reviewer Discussion:**

The reviewers brought up a number of valid concerns about positioning wrt past work, baseline comparisons, and unsubstantiated claims. Moreover additional visualizations would help the paper as well. While many of these suggestions did improve the paper comparisons and the authors added many of these, the primary concern about a set of tasks + corresponding foundation model generated reward as a baseline should really be incorporated to make the argument for skill discovery watertight.

---

### Decision · Program_Chairs · 2025-01-22

Reject